# Word2Fun: Modelling Words as Functions for Diachronic Word Representation

**Benyou Wang**[1]**, Emanuele Di Buccio**[1,2]**, and Massimo Melucci**[1]
[1]Department of Information Engineering
[2]Department of Statistical Sciences
University of Padova, Padova, Italy
{wang,dibuccio,melo}@dei.unipd.it

## Abstract

Word meaning may change over time as a reflection of changes in human society. Therefore, modeling time in word representation is necessary for some diachronic tasks. Most existing diachronic word representation approaches train the embeddings separately for each pre-grouped time-stamped corpus and align these embeddings, e.g., by orthogonal projections, vector initialization, temporal referencing, and compass. However, not only does word meaning change in a short time, word meaning may also be subject to evolution over long timespans, thus resulting in a unified continuous process. A recent approach called 'Diff-Time' models semantic evolution as functions parameterized by multiple-layer nonlinear neural networks over time. In this paper, we will carry on this line of work by learning explicit functions over time for each word. Our approach, called 'Word2Fun', reduces the space complexity from $\mathcal{O}(TVD)$ to $\mathcal{O}(kVD)$ where $k$ is a small constant ($k \ll T$). In particular, a specific instance based on polynomial functions could provably approximate any function modeling word evolution with a given negligible error thanks to the Weierstrass Approximation Theorem. The effectiveness of the proposed approach is evaluated in diverse tasks including time-aware word clustering, temporal analogy, and semantic change detection. Code at: https://github.com/wabyking/Word2Fun.git.

## 1   Introduction

Word meaning changes over time as a reflection of the changes of human society. Not only does word meaning change in a short time, word meaning may also be subject to evolution over long timespans. The causes of the change of word meaning may be cultural, societal, or technological [23]. Such lexical semantic change phenomenon has been investigated for some decades [3, 23].

Modeling the change of word meaning recently relies on distributed representations of words [2, 14] and adopting time-specific word vectors as diachronic word representation. The training is based on diachronic text corpora, which are obtained by partitioning the corpora into bins of textual documents labeled by time. A popular training paradigm within such an approach is known as *train-and-align*: first, the embeddings are trained separately for each pre-grouped time-stamped corpus; then, the embeddings are aligned by computing, e.g. orthogonal projections [9, 12], vector initialization [11], temporal referencing [7], parameter regularizers [26, 20], aligned compass [4], or latent diffusion [1]. The methods above do not model word meaning evolution as a continuous process, but consider one-hop transformation between pair of timestamps. However, the meaning of some words may smoothly evolve across a long-range span, according to a mixture of short and long timespans or in general according to unknown patterns which cannot straightforwardly be represented only considering one-hop transformations.

35th Conference on Neural Information Processing Systems (NeurIPS 2021).

A recent approach called 'DiffTime' [18] models semantic evolution as functions parameterized by multiple-layer nonlinear neural networks over time. Our paper carries on this line of work by modeling the change of word meaning as implicit functions for diachronic word representation, and considering word meaning evolution as a continuous process. These functions have considerable expressive power since polynomials can approximate any functions with a given negligible error thanks to the Weierstrass Approximation Theorem. Even if the proposed approach is general, we will focus on trigonometric functions.

We experimentally show that modelling each word as a finite number of trigonometric functions combined together in trigonometric polynomials is an effective approach to modeling the change of word meaning. Our experimental investigation also shows that the proposed method, even using fewer parameters, achieves better performance than the state-of-the-art (SOTA) diachronic word representation in time-aware word clustering, temporal analogy, and semantic change detection tasks, showing its potential for the research in diachronic word meaning representation and processing.

## 2 Background

### 2.1 Problem Definition

Representing words in vector space, i.e., a static word embedding [14] $h : \mathbb{N} \to \mathbb{R}^D$, is a common practice. However, static word embedding cannot model word dynamics over time, which is crucial when considering time-dependent corpora and some tasks, such as those carried out by specialists like linguists or sociologists. Diachronic word embedding (DWE) [9] (sometimes called Dynamic Word Embedding [1]) assumes that each word has a meaning which may vary with time. One word $w_i$ with index $i$ has its meaning represented as a $D$-dimensional vector $\mathbf{U}_{i,t} \in \mathbb{R}^D$ at a given time $t$. Therefore, DWE can be formalized as a mapping from $\mathbb{N} \times \mathbb{N}$ to $\mathbb{R}^D$, i.e. from the pair $(i, t)$ to a $D$-dimensional vector defined over a real field.

Most methods for DWE separately train embeddings for each time $t \in \mathbb{N}$. Note that even if we assume $t \in \mathbb{N}$, all formalizations can be extended to $t \in \mathbb{R}$; in practice, the granularity between two consecutive times is fixed, e.g., one year, thus representing $t$ in $\mathbb{N}$ is enough. The vectors that represent $V$ words in time $t$ are defined as $\mathbf{U}_{\cdot,t} \stackrel{def}{=} [\mathbf{U}_{1,t}, \mathbf{U}_{2,t} \cdots, \mathbf{U}_{V,t}] \in \mathbb{R}^{VD}$.

### 2.2 Alignment Issues for Rotation-invariant Embeddings

There are typically two ways to obtain word vectors: one method is matrix factorization, the other method is prediction-based neural methods; the former method is also related to the latter [13]. When considering the matrix factorization case, a common approach [26] is to approximate a given Positive Pointwise Mutual Information (PPMI) matrix $\mathbf{M}_t \in \mathbb{R}^{V \times V}$ by means of the $V$ word vectors $\mathbf{U}_{\cdot,t}$ such that $\mathbf{U}_{\cdot,t}\mathbf{U}_{\cdot,t}^T \approx \mathbf{M}_t$. The entries of $\mathbf{M}_t$ are measures of a relationship between words; for example, the entry at row $i$ and column $j$ may measure the degree of co-occurrence of the $i$-th word with the $j$-word within fixed-size textual windows. The word embeddings can be obtained by either using an eigenvalue method or a matrix factorization method which finds

$$\mathbf{U}_{\cdot,t} = \arg \min_{\mathbf{V}} |\mathbf{V}\mathbf{V}^T - \mathbf{M}_t|_F \tag{1}$$

However, we have that $\mathbf{U}'_{\cdot,t} = \mathbf{U}_{\cdot,t}\mathbf{R}$ can be an alternative solution of Eq. 1 for any orthogonal rotation matrix $\mathbf{R} \in \mathbb{R}^{D \times D}$. Indeed, $\mathbf{U}_{\cdot,t}\mathbf{R}(\mathbf{U}_{\cdot,t}\mathbf{R})^T = \mathbf{U}_{\cdot,t}\mathbf{R}\mathbf{R}^T(\mathbf{U}_{\cdot,t})^T = \mathbf{U}_{\cdot,t}(\mathbf{U}_{\cdot,t})^T$, since $\mathbf{R}\mathbf{R}^T = \mathbf{I}$. Therefore, training word embeddings according to Eq. 1 is non-convex [1] and as a consequence a global minimum might not exist.

Moreover, if diachronic word embeddings are trained twice on the same time-specific corpus yet with different random seeds, different solutions may result. It is also possible that two word vectors corresponding to different times, i.e., $\mathbf{U}_{\cdot,t_1}$ and $\mathbf{U}_{\cdot,t_2}$ cannot be compared because they might be arbitrarily rotated. One has to align these time-specific word embeddings. The rotation-invariant issue is also present for prediction-based word embeddings e.g., Word2Vec [14].

Table 1: Comparison of existing diachronic word representation and Word2Fun. $i$ refers to a word index; $j$ refers to a dimension index. The target embedding $\boldsymbol{C} \in \mathbb{R}^{VD}$ is not considered in this table.

| Diachronic word vectors | | formalization | Param. |
|---|---|---|---|
| Word2Vec | static word embeddings | $\boldsymbol{W}_i$ | $VD$ |
| Aligned Diachronic Word2Vec | linear projection [12, 9] vector initialization [11] discrepancy penalty [20, 26] latent diffusion [1] temporal referencing [7] aligned by compass [4] | $\mathbf{B}_{i,t}$ ; $\mathbf{B}_{i,t} \in \mathbb{R}^D$ | $TVD$ |
| | DiffTime [18] | NN $(f_{\text{word}}(i), f_{\text{time}}(t))$, NN | $2VD + D^3 + 4D^2$ |
| Word2Fun | linear function | $\boldsymbol{W}_i + \boldsymbol{K}_i t$ | $2VD$ |
| | I (Time2Fun ) | $\boldsymbol{B}_i + \sin(\boldsymbol{\Omega} t)$ | $VD + D$ |
| | II (w/ fixed freqs ) | $\boldsymbol{B}_i + \boldsymbol{R}_i[\sin(\boldsymbol{\Omega} t; \cos(\boldsymbol{\Omega} t)], \boldsymbol{\Omega}_j = \frac{1}{10000}^{j/\frac{D}{2}}$ | $2VD$ |
| | III (w/ trainable freqs ) | $\boldsymbol{B}_i + \boldsymbol{R}_i[\sin(\boldsymbol{\Omega}_i^{(1)} t); \cos(\boldsymbol{\Omega}_i^{(2)} t)]$ | $3VD$ |
| | IV (w/ trainable phases) | $\boldsymbol{B}_i + \boldsymbol{R}_i[\sin(\boldsymbol{\Omega}_i^{(1)} t + \boldsymbol{\Theta}_i^{(1)}); \cos(\boldsymbol{\Omega}_i^{(2)} t + \boldsymbol{\Theta}_i^{(2)})]$ | $4VD$ |

## 2.3 Existing Work of Diachronic Word Embedding

Existing DWE, e.g., [1, 9] considers word meaning evolution in two consecutive time-specific corpora (e.g., $t_1$ and $t_2$), instead of a whole continuous process. They train word vectors for each time-specific corpus and then *align* them using orthogonal projections [9, 12], vector initialization [11], temporal referencing [7], parameter regularizers [26, 20], aligned compass [4], latent diffusion [1]. Tab. 1 summarizes these approaches. The number of parameters of the alignment-based dynamic word embeddings is proportional to the number $T$ of time bins, resulting in a parameter space of $O(TVD)$. This is a notable cost with a long time span, e.g., splitting COHA dataset yearly results in 200 time slices ($T = 200$). Furthermore, transformations between two any consecutive time bins are independent, namely, $\mathbf{U}_{i,t_1}, \mathbf{U}_{i,t_2}$ may be totally independent of the one between $\mathbf{U}_{i,t_2}, \mathbf{U}_{i,t_3}$ even if they are close in the time. Note that these approaches might lose some power when modeling such sequential evolution with more than two time bins in the event of word meaning that gradually evolves during a longer timespan, e.g., many decades. Lastly, dividing a corpus into separate time bins may lead to many smaller training sets. Word vectors in a time bin are trained with roughly $\frac{1}{T}$ training examples instead of all examples. This might be an issue with time-unbalanced dynamic corpora. For example, when there are much fewer documents in a specific year $t$ than others, the word-specific word vectors in that year may be distorted because of underfitting; it may therefore negatively affect the long-range transformation of word meaning change.

DiffTime [18] was the first work to model word meaning evolution as a continuous process to the best of our knowledge. It leverages multiple-layer neural networks to aggregate output from a word encoder $f$ and a time encoder $g$. Even if the parameter scale does not increase with a bigger $T$, DiffTime additionally introduces much more parameters (e.g., $D^3$). Such learned functions are less interpretable due to the depth and non-linearity of neural networks. We argue that the neural network architecture in [18] is ad hoc without theoretical approximation guarantee for semantic shift.

## 3 Methodology: Modelling Words as Functions

### 3.1 Formulation

To smoothly model word meaning change over time we represent each word as a continuous function: a specific word vector at time $t$ is represented as the values of the function when the variable equals $t$, which is inspired by [24, 18, 10, 25]. More formally, our approach aims to learn a mapping $f$ that maps each word $w_i$, with index $i \in \mathbb{N}$, to functions over time:

$$f : \mathbb{N} \to (g : \mathbb{N} \to \mathbb{R}^D)$$

where $g$ is a function over a variable $t \in \mathbb{N}$. Note that the output of $g$ is a $D$-dimensional vector, $g(t) \in \mathbb{R}^D$. Let us denote $f(i)$ as $g_i$. A word $w_i$ at time $t$ is represented as a $D$-dimensional vector $\mathbf{U}_{i,t} = f(i)(t) = g_i(t)$.

Examples of $g$ are linear functions $g(t) = \boldsymbol{b} + \boldsymbol{k}t$ with parameters $\boldsymbol{b}, \boldsymbol{k} \in \mathbb{R}^D$ or a sinusoidal functions $g(t) = \boldsymbol{b} + \boldsymbol{r}\sin(\boldsymbol{\omega}t + \boldsymbol{\theta})$ with parameters $\boldsymbol{b}, \boldsymbol{r}, \boldsymbol{\omega}, \boldsymbol{\theta} \in \mathbb{R}^D$. Section 4 reports some remarks on the considered instantiations of $g$ and how they are linked to previously proposed approaches.

## 3.2 A Skip-gram Implementation with a Compass

The Skip-gram model architecture tries to predict the source context words (surrounding words) given a target word (the center word), which was considered to achieve the reverse of what the CBOW model does. Let us denote $u$ as the target word and $v$ as the source word, and negatively-sampled target words $\hat{\mathbb{V}} = \{\hat{v}_i\}$. It learns a diachronic word representation method $f : \mathbb{N} \times \mathbb{R} \to \mathbb{R}^D$ and a static word mapping $h : \mathbb{N} \to \mathbb{R}^D$; $f(u, t)$ denotes the diachronic word vector of $w_u$ in time $t$.

The static word embedding $h(u)$ is introduced to align the diachronic word vectors in different times; it is called a *compass* in [4]. The compass is used as an anchor to project time-specific word embedding in different times to unified designated vector space. Our implementation of the *compass* differs from [4], since we jointly train the static compass and diachronic word vectors simultaneously, thus avoiding a two-stage training and the compass can be non-fixed in the second stage.

The loss function is defined as:

$$\mathcal{L} = - \sum_{(u,v,\hat{\mathbb{V}},t) \in \mathbb{D}_{\text{train}}} \left( \log(\delta(f(u,t)h(v)^T)) + \frac{1}{|\hat{\mathbb{V}}|} \sum_{\hat{v}_i \in \hat{\mathbb{V}}} \log(\delta(-f(u,t)h(\hat{v}_i)^T)) \right) \tag{2}$$

$\delta$ is the sigmoid activation function. The algorithm is explained in Algo. 1, and the details on the parameterization are reported in Tab. 1.

---

**ALGORITHM 1:** Training algorithm for Word2Fun.

---

**Require:** time-stamped corpora $\mathcal{C}$;
         a diachronic word mapping $f : (\mathbb{N}, \mathbb{R}) \to \mathbb{N}^D$
         a static word mapping $h : \mathbb{N} \to \mathbb{R}^D$
 1: shuffling time-stamped corpora
 2: **for** each epoch **do**
 3:    **for** each document $d$ with a timestamp $t$ in $\mathcal{C}$ **do**
 4:      **for each** skip-gram $(u, v)$ in $d$ **do**
 5:         calculating the loss for the positive sample $\mathcal{L}_{\text{pos}} = \log(\delta(f(u,t)h(v)^T))$
 6:         generating negatively-sampled target words $\hat{\mathbb{V}}$
 7:         calculating the loss for negative samples $\mathcal{L}_{\text{neg}} = -\sum_{\hat{v}_i \in \hat{\mathbb{V}}} \log(\delta(-f(u,t)h(\hat{v}_i)^T))$
 8:         back-propagating using the loss $\mathcal{L} = \mathcal{L}_{\text{pos}} + \mathcal{L}_{\text{neg}}$
 9:      **end for**
10:    **end for**
11: **end for**

---

# 4 Which Functions?

## 4.1 Function Approximation using Polynomials

For a skip-gram pair $(w_i, w_j)$,[1] the similarity degree between a context word $w_i$ and a target word $w_j$ in time $t$ is calculated as a dot product between the time-specific context embedding $f(i, t)$ and the static target embedding $h(j)$, the latter being the 'compass':

$$y_{i,j}(t) = f(i,t)h(j)^T \tag{3}$$

$y_{i,j}(t)$ is function of time and it measures the *similarity between $w_i$ and $w_j$ over time*;[2] we argue that semantic meaning evolution can be captured by approximating arbitrary between-word similarity over time. How to approximate $y_{i,j}(t)$ by selecting appropriate parameterization of $f$ and $h$?

---

[1] If $w_i$ and $w_j$ appear together in a $h$-size window, we call $(w_i, w_j)$ as a skip-gram pair in this paper, $w_i$ is the context word and $w_j$ is the target word, and vice versa.

[2] For PPMI factorization to obtain word vectors, $y_{i,j}(t)$ is the changing PPMI between $w_i$ and $w_j$ over time.

Let us recall the Weierstrass Approximation Theorem that states that every continuous real function defined on $[a, b]$ can be approximated by a real polynomial. Note that Theorem 1 also holds for trigonometric polynomial defined in Definition 1, see Corollary 1.

**Theorem 1.** *Weierstrass Approximation Theorem. Let $F$ be a continuous real-valued function defined on the real interval $[a, b]$. For any $\epsilon > 0$, there exists a polynomial $P$ such that for all $x \in [a, b]$, we have $|F(x) - P(x)|_\infty < \epsilon$.*

**Definition 1.** A **trigonometric polynomial** of degree $D$ is an expression of the form $\Delta + \sum_{k=1}^{D} \alpha_k \cos(kx) + \beta_k \sin(kx)$, where $\Delta, \alpha_1, \ldots, \alpha_D, \beta_1, \ldots, \beta_D \in \mathbb{R}$.

**Corollary 1.** *Approximation by trigonometric polynomials For every continuous $2\pi$-periodical function $F : \mathbb{R} \to \mathbb{R}$ defined on the real interval $[0, 2\pi]$, and for any $\epsilon > 0$, there exists a trigonometric polynomial $P$ such that for all $x \in [0, 2\pi]$, we have $|F(x) - P(x)|_\infty < \epsilon$.*

**Corollary 2.** *The trigonometric polynomials $span\{1, \sin x, \cos x, \cdots, \sin(Dx), \cos(Dx)\}$ is dense in $\mathcal{C}[0, 2\pi]$ iff $span\{1, \sin \frac{2\pi x}{a-b}, \cos \frac{2\pi x}{a-b}, \cdots, \sin \frac{2\pi Dx}{a-b}, \cos \frac{2\pi Dx}{a-b}\}$ is dense in $\mathcal{C}[a, b]$.*

From Corollary 1 [16, 27] we know that a trigonometric polynomial $\{0, \sin x, \cos x, \cdots, \sin(Dx), \cos(Dx)\}$ spans a subspace that can approximate any periodical continuous functions defined on $[0, 2\pi]$, i.e., it is dense in $\mathcal{C}[0, 2\pi]$. By Corollary 2, one can conclude that a weighted sum of trigonometric functions with appropriate periods can approximate any periodical continuous functions defined in an arbitrary closed interval [16].

## 4.2 Sinusoidal Parameterization in Word2Fun

Since the static embedding $h(j)$ is not related to time $t$, we consider $f(i, t)$ with sinusoidal paramerization (a mixture of cosine and sine functions plus a bias term). Then Eq. 3 will result in:[3]

$$
\begin{aligned}
y_{i,j}(t) = f(i,t)h(j)^T = \sum \left( \begin{bmatrix} \boldsymbol{B}_{i,1} + \boldsymbol{R}_{i,1}\sin(\Omega_1 t) \\ \boldsymbol{B}_{i,2} + \boldsymbol{R}_{i,2}\cos(\Omega_1 t) \\ \cdots \\ \boldsymbol{B}_{i,D-1} + \boldsymbol{R}_{i,D-1}\sin(\Omega_{\frac{D}{2}} t) \\ \boldsymbol{B}_{i,D} + \boldsymbol{R}_{i,D}\cos(\Omega_{\frac{D}{2}} t) \end{bmatrix} \odot \begin{bmatrix} \boldsymbol{C}_{j,1} \\ \boldsymbol{C}_{j,2} \\ \cdots \\ \boldsymbol{C}_{j,D-1} \\ \boldsymbol{C}_{j,D} \end{bmatrix} \right) \\
= \underbrace{\sum_{k=1}^{D} \boldsymbol{B}_{i,k}\boldsymbol{C}_{j,k}}_{\Delta} + \sum_{k=1}^{\frac{D}{2}} \underbrace{\boldsymbol{R}_{i,2k-1}\boldsymbol{C}_{j,2k-1}}_{\alpha_{i,j,k}}\sin(\Omega_k t) + \underbrace{\boldsymbol{R}_{i,2k}\boldsymbol{C}_{j,2k}}_{\beta_{i,j,k}}\cos(\Omega_k t)
\end{aligned}
\tag{4}
$$

Therefore, $y_{i,j}(t)$ is a weighted sum of sinusoidal functions plus a constant term $\Delta = \sum_{k=1}^{D} \boldsymbol{B}_{i,k}\boldsymbol{C}_{j,k}$. By replacing the coefficients with $\alpha_{i,j,k}$ and $\beta_{i,j,k}$, we can rewrite Eq. 4 as $y_{i,j}(t) = \Delta + \sum_{k=1}^{\frac{D}{2}} \alpha_{i,j,k}\sin(\Omega_k t) + \beta_{i,j,k}\cos(\Omega_k t)$; $\{\alpha_{i,j,k}\}_{k=1}^{\frac{D}{2}}$ and $\{\beta_{i,j,k}\}_{k=1}^{\frac{D}{2}}$ are the coefficients and $\{\Omega_k\}_{k=1}^{\frac{D}{2}}$ are the corresponding frequencies. Following the argument in [5] when discussing approximation properties of sine and cosine activation functions, since linear combinations of sine and cosine generate all finite trigonometric polynomials that have approximation properties described in Section 4.1, the dot product between the diachronic embedding $f(i, t)$ and the static target embedding $h(i)$, denoted as $y_{i,j}(t)$ can capture any evolving relations between arbitrary skip-gram pairs.

By iteratively training all skip-gram pairs, Word2Fun can model complicated semantic change across time. Intuitively, small frequencies would reflect some long-range evolution, while some big frequencies would capture short-range evolution. Such periodical property would allow such functions to capture long enough evolution without considering boundedness issues.

## 4.3 Investigated Function Parameterizations

**Linear functions.** A real polynomial naturally induce: $\mathbf{U}_{i,t} = g_i(t) = \boldsymbol{b}_i + \boldsymbol{k}_i \odot [1, t, t^2, \cdots t^D]^T$. Since $t^D$ is computationally expensive and sometimes meets the overflow issue, we choose a simpler case with a linear function parameterization of Word2fun: $\mathbf{U}_{i,t} = g_i(t) = \boldsymbol{b}_i + \boldsymbol{k}_i t$.

---

[3]$\odot$ is the element-wise multiplication

Table 2: Statistics of diachronic corpora.

| corpus | num. of tokens | time range | time granularity | T | Vocab |
|---|---|---|---|---|---|
| COHA [6] | 472M | 1810 - 2009 | every decade | 20 | 43, 734 |
| New York Times [26] | 105M | 1990 - 2016 | yearly | 27 | 20, 314 |

**Sinusoidal functions.** In this work, we investigate the sinusoidal parameterization. We use a combination of sine and cosine functions; i.e., half of dimensions are parameterized by sine functions and half of them are parameterized by cosine functions. Considering the domain of time $t \in \mathbb{N}$ and its range in this paper, we adopt a set of frequencies $\{\boldsymbol{\omega}_i\}$ [24] that is different from standard trigonometric polynomial. The *Word2Fun* II is defined as below:

$$g_i(t) = \boldsymbol{b}_i + \boldsymbol{r}_i[\cos(\boldsymbol{\Omega}t); \sin(\boldsymbol{\Omega}t)], \quad \boldsymbol{\Omega}_j = \frac{1}{10000}^{j/\frac{D}{2}}, \ j \in [1, \cdots, \frac{D}{2}] \quad (5)$$

Note we add a static component $\mathbf{b}_i$ that is independent from time. In *Word2Fun* III we also learn word-specific frequencies: $\mathbf{U}_{i,t} = g_i(t) = \boldsymbol{b}_i + \boldsymbol{r}_i[\sin(\boldsymbol{\omega}_i^{(1)}t); \cos(\boldsymbol{\omega}_i^{(2)}t)]$ [4]. *Word2Fun* IV adds trainable word-specific phases: $\mathbf{U}_{i,t} = g_i(t) = \boldsymbol{b}_i + \boldsymbol{r}_i[\sin(\boldsymbol{\omega}_i^{(1)}t + \boldsymbol{\theta}_i^{(1)}); \cos(\boldsymbol{\omega}_i^{(2)}t + \boldsymbol{\theta}_i^{(2)})]$, see Tab. 1. $\boldsymbol{\omega}_i^{(1)} \in \mathbb{R}^{\frac{D}{2}}$ is different from $\boldsymbol{\omega}_i^{(2)} \in \mathbb{R}^{\frac{D}{2}}$. $\boldsymbol{\theta}_i^{(1)} \in \mathbb{R}^{\frac{D}{2}}$ is different from $\boldsymbol{\theta}_i^{(2)} \in \mathbb{R}^{\frac{D}{2}}$ .

Standard static word vectors, e.g., [14], can be considered as constant functions, a special case of Word2fun since $\boldsymbol{b}_i$ is a time-agnostic word vector. For instance, (1) in the case of the linear function, $\boldsymbol{k}_i = \mathbf{0}$; (2) in the case of sinusoidal function, $\boldsymbol{r}_i = \mathbf{0}$ or $\boldsymbol{\omega}_i$ is small enough. Both these cases results in $g_i = \boldsymbol{b}_i$. The additional parameters $\boldsymbol{\omega_i}$ and $\boldsymbol{r}_i$ are expected to capture the word meaning evolution.

**Time2Fun: purely encoding time as functions.** We can separately learn a word mapping and a time mapping:

$$f_{\text{word}} : \mathbb{N} \longrightarrow \mathbb{R}^D, \quad f_{\text{time}} : \mathbb{N} \longrightarrow \mathbb{R}^D \quad (6)$$

Then $w_i$ at time $t$ can be represented as $\mathbf{U}_{i,t} = f_{\text{word}}(i) + f_{\text{time}}(t)$, where $f_{\text{time}}$ can be a function over time, e.g., $f_{\text{time}} = \boldsymbol{K}t$ or $f_{\text{time}} = \sin(\boldsymbol{\Omega}t)$ — the latter corresponds to Word2Fun I in Tab. 1. Time2Fun is close to [18] which represents time as a continuous variable and model a word's usage as a function of time. [10] also explores to use time as an extra feature for various tasks.

# 5 Experiments

## 5.1 Data and Experimental Settings

**Corpora.** The diachronic corpora used in this paper are reported in Tab. 2. The Corpus of Historical American English (**COHA**) [6] is the largest structured corpus of historical English (the 1810s-2010s), contains more than 475 million words and is balanced by genre decade by decade. The **New York Times** (NYT) [26] contains 99,872 articles published between January 1990 and July 2016; besides the article text, metadata including title, author, release date, and section label were also collected.

**Baselines.** We choose the baselines from [26]. *Static Word2Vec*: the standard word2vec embeddings [14], trained on the entire corpus and ignoring time information. *Transformed Word2Vec* [12]: the embeddings are first trained separately by factorizing PPMI matrix for each year, and then transformed by optimizing a linear transformation matrix which minimizes the distance between two consequent time-stamped trained embeddings for the k nearest words' embeddings to the querying words. *Aligned Word2Vec* [9] : the embeddings are first trained by factorizing the PPMI matrix for each year $t$, and then aligned by searching for the best orthonormal transformation between two consequent time-stamped trained embeddings. *Dynamic Word Embedding* [26] learns word embedding by factorizing PPMI matrix in individual time; plus, it imposes a regularizer to encourage two subsequent word embedding being similar for alignment. Besides those baselines, we considered DiffTime [18] (described in Sec. 2.3) and *Compass Aligned Word Embedding* [4] that exploits fixed target embeddings to align time-specific context embeddings. The former has a embedding size of 150 and its hidden dimension is 50, and the latter has a embedding size of 100; the both have more parameters than Word2Fun.

---

[4]Sinusoidal functions introduce infinite local minima due to the periodicity of sinusoidal functions [15], but we empirically found it performs well [10] and leave the above issue a future work.

Table 3: Experimental results of time-aware word clustering.

| Method | 10 Clusters | | 15 Clusters | | 20 Clusters | |
|---|---|---|---|---|---|---|
| | NMI | $F_\beta$ | NMI | $F_\beta$ | NMI | $F_\beta$ |
| Global/static word vector [14] | 0.6736 | 0.6163 | 0.6867 | 0.7147 | 0.6713 | 0.7214 |
| Transformed Word2Vec [12] | 0.5175 | 0.4584 | 0.5221 | 0.5072 | 0.5130 | 0.5373 |
| Aligned Word2Vec [9] | 0.6580 | 0.6530 | 0.6618 | 0.7115 | 0.6386 | 0.7187 |
| Dynamic Word2Vec [26] | 0.7175 | 0.6949 | 0.7162 | 0.7515 | 0.6906 | 0.7585 |
| Compass aligned Word2Vec [4] | 0.5191 | 0.3750 | 0.5062 | 0.4051 | 0.5077 | 0.4331 |
| DiffTime [18] | 0.6945 | 0.6723 | 0.6727 | 0.7161 | 0.6576 | 0.7275 |
| Word2Fun linear | 0.1676 | 0.1813 | 0.2826 | 0.3035 | 0.2473 | 0.2932 |
| Word2Fun I (Time2Fun) | 0.1703 | 0.1783 | 0.2691 | 0.2680 | 0.2842 | 0.2649 |
| Word2Fun II | **0.7281** | **0.7147** | **0.7181** | 0.7645 | **0.7012** | 0.7616 |
| Word2Fun III | 0.7233 | 0.7080 | 0.7086 | **0.7701** | 0.6980 | **0.7630** |
| Word2Fun IV | 0.7111 | 0.6913 | 0.7023 | 0.7451 | 0.6823 | 0.7602 |

**Experimental settings.** We removed words that appear less than 200 times, as [26] did. We used the Adam optimizer with a learning rate of 0.0025. The batch size can be as big as it achieves the upper bound of memory of GPU. The model converged in less than 20 epochs with no drop in terms of loss. We computed the average performance on the last three saved checkpoints. For sinusoidal functions, we used a mixture of cosine functions and sine functions. The dimension was 100; we preliminary found that models with bigger dimensions (e.g. 200 or 300) can slightly improve performance. Word2Fun with all settings was slightly slower than the standard Word2Vec due to additional computation for obtaining word vectors. However, all methods include Word2Vec and Word2Fun took 10-15 minutes for one epoch in a single Nvidia V100 32G GPU for NYT dataset.

## 5.2 Quantitative Evaluation

### 5.2.1 Time-aware Word Clustering

The time-aware word clustering task was proposed in [26] and exploits the NYT dataset. If one word was extremely frequent in a particular section, it was labeled as most-used in that section.[5] Such section annotation was used to evaluate the clustering results. Across 11 sections, there were 1,888 triplets denoted as $(w_i, t_i, s_i)$ to indicate that word $w_i$ in year $t_i$ was associated with section $s_i$.

We apply spherical $K$-means on learned time-specific word vectors using cosine distance, with $K$ = 10, 15, and 20 clusters. As in [26], we used *Normalized Mutual Information* (NMI) and $F_\beta$-*measure* to measure the consistency between section label and clustering results. NMI was defined as $\text{NMI}(\mathcal{C}, \mathcal{S}) = 2\text{MI}(\mathcal{C}, \mathcal{S})/(\text{E}(\mathcal{C}) + \text{E}(\mathcal{S}))$, where $\mathcal{S}$ denotes section labels ($\{s_i\}$) for word-year pairs $\{(w_i, t_i)\}$ and $\mathcal{C}$ denotes clustering categories ($\{c_i\}$) for these word-year pairs $\{(w_i, t_i)\}$. $\text{MI}(\cdot, \cdot)$ and $\text{E}(\cdot)$ are the functions to compute Mutual Information and Entropy respectively. $F_\beta = \frac{(\beta^2+1)PR}{\beta^2 P + R}$ measures the effectiveness as a $\beta$-weighted harmonic mean of precision and recall. As in [26], we set $\beta = 5$ to emphasise recall to penalizing false negative more.

**Experimental results.** As shown in Tab. 3, the proposed Word2fun II, III and IV outperformed all the baseline methods. Word2Fun with linear parameterization and Word2fun I performed nearly the worst, since they could not learn word-dependent evolution. Word2fun IV achieved worse performance than III, thus showing that adding phase seems unbeneficial to Word2Fun.

### 5.2.2 Temporal Analogy

The Temporal Analogy task [22] utilizes quadruples $(w^{(1)}, t^{(1)}, w^{(2)}, t^{(2)})$ to say that "$w^{(1)}$ in year $t^{(1)}$" is like "$w^{(2)}$ in year $t^{(2)}$". To examine the quality of temporal analogy, [26] created a task to investigate equivalences across years. For example, given *obama* in *2012*, we aim to find its equivalent word in 2002. As we know *obama* was the U.S. president in 2012, its equivalent word

---

[5]News articles in NYT dataset are tagged with their 'sections' such as 'Business', 'Sports', 'World', and 'Technology'. For example, we see that *amazon* occurs 41% of the time in *World* in 1995, associating strongly with the rainforest, and 50% of the time in *Technology* in 2012, associating strongly with *e-commerce*.

Table 4: Experimental results of temporal analogy in *test1*

| Method | MRR | P@1 | P@3 | P@5 | P@10 |
|---|---|---|---|---|---|
| Global/static Word2Vec [14] | 0.3560 | 0.2664 | 0.4210 | 0.4774 | 0.5612 |
| Transformed Word2Vec [12] | 0.0920 | 0.0500 | 0.1168 | 0.1482 | 0.1910 |
| Aligned Word2Vec [9] | 0.1582 | 0.1066 | 0.1814 | 0.2241 | 0.2953 |
| Dynamic Word2Vec [26] | 0.4222 | 0.3306 | 0.4854 | 0.5488 | 0.6191 |
| Compass aligned Word2Vec [4] | **0.481** | **0.404** | **0.534** | 0.582 | 0.636 |
| DiffTime [18] | 0.3759 | 0.2738 | 0.4200 | 0.5070 | 0.5912 |
| Word2Fun linear | 0.3016 | 0.2649 | 0.3255 | 0.3426 | 0.3630 |
| Word2Fun I (Time2Fun) | 0.3735 | 0.2646 | 0.4300 | 0.4955 | 0.5874 |
| Word2Fun II | 0.4061 | 0.2756 | 0.4916 | 0.5614 | 0.6434 |
| Word2Fun III | 0.4354 | 0.3076 | 0.5330 | **0.5837** | **0.6647** |
| Word2Fun IV | 0.4208 | 0.2954 | 0.5076 | 0.5715 | 0.6470 |

Table 5: Experimental results of temporal analogy in *test2*

| Method | MRR | P@1 | P@3 | P@5 | P@10 |
|---|---|---|---|---|---|
| Global/static Word2Vec [14] | 0.0472 | 0.0000 | 0.0787 | 0.0787 | 0.2022 |
| Transformed Word2Vec [12] | 0.0664 | 0.0404 | 0.0764 | 0.0989 | 0.1438 |
| Aligned Word2Vec [9] | 0.0500 | 0.0225 | 0.0517 | 0.0787 | 0.1416 |
| Dynamic Word2Vec [26] | 0.1444 | 0.0764 | 0.1596 | 0.2202 | 0.3820 |
| Compass Aligned Word Embedding [4] | 0.1361 | 0.0749 | 0.1918 | 0.2904 | 0.3918 |
| DiffTime [18] | 0.1259 | 0.0055 | 0.1562 | 0.2219 | 0.3973 |
| Word2Fun linear | 0.0425 | 0.0137 | 0.0384 | 0.0630 | 0.1014 |
| Word2Fun I (Time2Fun) | 0.0992 | 0.0000 | 0.1315 | 0.1726 | 0.2849 |
| Word2Fun II | 0.1194 | 0.0358 | 0.1075 | 0.2219 | 0.3863 |
| Word2Fun III | **0.1824** | **0.0795** | **0.1973** | **0.2932** | **0.4164** |
| Word2Fun IV | 0.1536 | 0.0548 | 0.1562 | 0.2411 | 0.3918 |

in 2002 is *bush*, who was the U.S. president in 2002. In this way, there are two test sets. The first set (test1) is based on publicly recorded knowledge that lists different names for a particular role, such as U.S. president for each year. Human experts generated the second test (test2) to explore emerging technologies, brands, and significant events (e.g., disease outbreaks and financial crisis), etc. Reciprocal Rank (MRR) and Precision@K (P@K) are used for the evaluation.

**Experimental results.** As shown in Tab. 4 and Tab. 5, Word2Fun III outperformed all baselines in *test2* and some metrics (including P@5, P@10) in *test1*. In *test1*, Word2Fun performed worse than Compass-aligned Word2Vec [4] in terms of MRR and P@1; this might be explained by Word2Fun has fewer parameters than [4].

### 5.2.3 Semantic Change Detection

Semeval-2020 task 'Unsupervised lexical semantic change detection' [21] released a dataset for Semantic Change Detection with human-annotated semantically-shift degrees. It provides semantically-shift degree of thirty seven English words from the timespan 1810-1860 to the timespan 1960-2010. The task aims to predict semantically-shift degree and the performance is evaluated by Pearson and Spearman correlation. We split data of every decade into time bins. We used Word2Fun trained on COHA dataset. For each word, we take the cosine distance between the average word vectors in the first five decades and last five decades as the semantic change degree.

**Experimental Results.** As shown in Tab. 6, Word2Fun outperformed the five mentioned baselines and the most effective approaches in the Semeval-2020 task. Note that the task participants [19, 17] did not train on the same text collection; they used a subset of COHA and we use a complete one.

### 5.3 Analysis of Word2Fun

**Parameter interpretability** To better understand the learned parameters, we will consider Word2Fun III since it was the best performing one. In Word2Fun III, a word $w_i$ is represented

Table 6: Semantic change detection. Baselines in the first group are implemented by this work.

| models | Pearson | Spearman |
|---|---|---|
| Global/static Word2Vec [14] | nan | nan |
| Transformed Word2Vec [12] | 0.0727 | 0.0865 |
| Aligned Word2Vec [9] | 0.3333 | 0.3083 |
| Dynamic Word2Vec [26] | 0.2727 | 0.2877 |
| Compass aligned word embedding [4] | 0.3199 | 0.2567 |
| DiffTime [18] | 0.3504 | 0.4146 |
| Word2Fun linear | -0.1200 | -0.0790 |
| Word2Fun I (Time2Fun) | 0.3925 | 0.4550 |
| Word2Fun II | 0.4478 | **0.5038** |
| Word2Fun III | **0.5355** | 0.4057 |
| Word2Fun IV | 0.4483 | 0.3578 |
| multilingual BERT [19] (SemEval-2020 1st) | - | 0.436 |
| ensemble between aligned Word2Vec and BERT [17] (SemEval-2020 2nd) | - | 0.422 |

Table 7: Learned parameters for semantic change detection. AVG means average for a vector.

| Learned parameters | Definition | Pearson | Spearman |
|---|---|---|---|
| frequencies term | $AVG(|\boldsymbol{\Omega_i}|)$ | 0.0958 | 0.1748 |
| amplitude term | $AVG(|\boldsymbol{R_i}|)$ | 0.1213 | 0.0474 |
| bias term | $AVG(|\boldsymbol{B_i}|)$ | -0.3640 | **-0.2636** |
| ratio between the amplitude and bias terms | $AVG(|\boldsymbol{R_i}|)/AVG(|\boldsymbol{B_i}|)$ | **0.4141** | 0.2358 |

as $\boldsymbol{B}_i + \boldsymbol{R}_i[\sin(\boldsymbol{\Omega_i}^{(1)}t); \cos(\boldsymbol{\Omega_i}^{(2)}t)]$ where $\boldsymbol{B}_i, \boldsymbol{R}_i, \boldsymbol{\Omega_i} \in \mathbb{R}^D$. There are three cases where the function degrade to a constant function: a) Absolute values of elements in $\boldsymbol{B}_i$ is much larger than that of $\boldsymbol{R}_i$; b) $\boldsymbol{R}_i = \boldsymbol{0}$; or c) $\boldsymbol{\Omega_i} = \boldsymbol{0}$. Intuitively, smaller $\boldsymbol{B}_i$, bigger $\boldsymbol{R}_i$ or $\boldsymbol{\Omega_i}$ indicates that the word $w_i$ is sensitive to time. In other words, *the meaning of words more likely changes over time in the case of smaller $\boldsymbol{B}_i$, bigger $\boldsymbol{R}_i$ or $\boldsymbol{\Omega_i}$*.

To examine the above intuition, we define the average of the absolute values for these parameters (i.e., $\boldsymbol{B}_i, \boldsymbol{R}_i, \boldsymbol{\Omega_i}$) computed over all the dimensions as indicators for semantically-shift degree. Since in Word2Fun III, $\boldsymbol{B}_i$ is the time-unrelated term while $\boldsymbol{R}_i[\sin(\boldsymbol{\Omega_i}^{(1)}t); \cos(\boldsymbol{\Omega_i}^{(2)}t)]$ is the time related term, we also take the ratio between $AVG(|\boldsymbol{R}_i|)$ and $AVG(|\boldsymbol{B}_i|)$ as an indicator (last row of Tab. 7). Tab. 7 confirms our intuition. Especially, the Pearson correlation values of last two rows are significant with $p < 0.05$. Although the result in Tab. 7 is not as good as the results in Tab. 6, Tab. 7 is surprising since these indicators do not consider the two specific evaluation timespans and they therefore capture the word evolution "speed" over the whole timespan.

When checking distribution of these parameters, we observe that semantically-shifted words generally have slightly bigger frequencies, bigger amplitudes, and smaller biases. Interestingly, nearly 70% of the sinusoidal functions does not complete a whole period in 20 decades (from 1810s to 2010s).

**Visualization of learned functions** Learned functions for the top 4 semantically-shifted words and another top 4 semantically-unshifted words[6] are visualized in Fig. 1: we selected two dimensions for cosine functions (see Fig. 1a and 1b) and two dimensions for sine functions (see Fig. 1c and 1d) in Word2Fun III. Fig. 1a shows that the function in first dimension learns nearly constant functions for the two semantically-unshifted words (namely, chairman and relationship), indicating the two words will not change their representation over time for this dimension. In Fig. 1b, we could observe that amplitudes for semantic-shifted words are bigger than amplitudes for semantic-unshifted words. In Fig. 1c and 1d with sine functions, the evolution for semantic-shifted words seems to be severe than semantic-unshifted words. In Fig. 1c, it shows some *long-term evolution* with bigger periods (smaller frequencies). We argue that the proposed Word2Fun has the potential to model long-term word meaning evolution, such long-term evolution might span decades or even centuries [8] especially for general language usage shift.

---

[6]We used the annotated test set in Sec. 5.2.3. The top 4 semantically-shifted words are graft, plane, prop, tip; the top 4 semantically-unshifted words are chairman, fiction, relationship, risk.

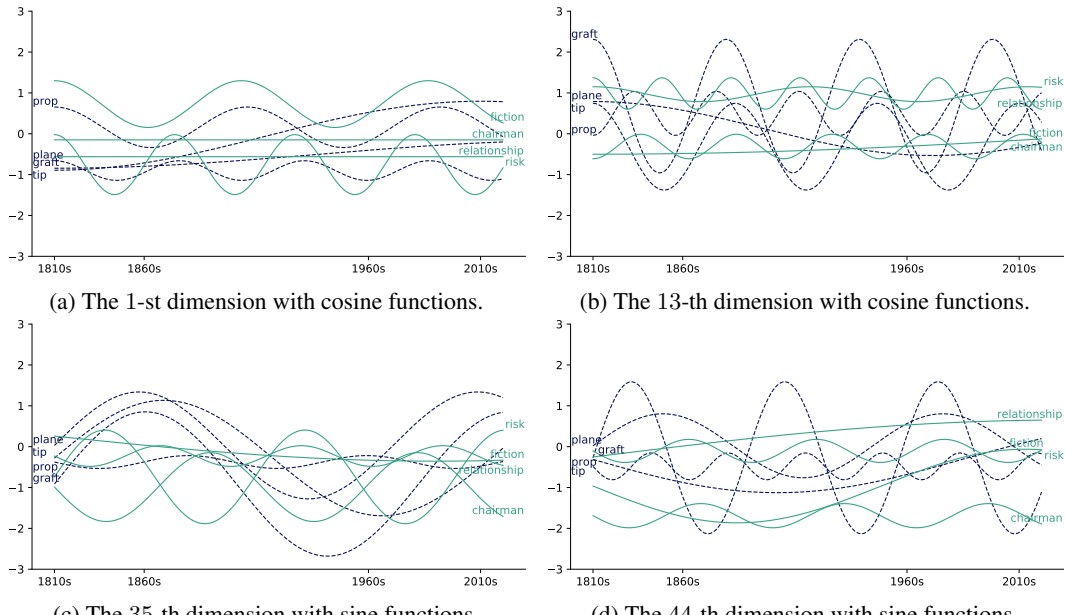

(a) The 1-st dimension with cosine functions.

(b) The 13-th dimension with cosine functions.

(c) The 35-th dimension with sine functions.

(d) The 44-th dimension with sine functions.

Figure 1: We show some learned functions for top 4 semantically-shifted words in purple and top 4 semantically-unshifted words in blue, for Word2Fun III with 50 dimension. The first half dimensions (1-25) use cosine functions while the last half dimensions (26-50) use sine functions.

Table 8: Most similar words for *apple*, *european*, *phone*, and *browser* from the year 1990 (denoted as 90) to 2016 (denoted as 16). All words are lower cased.

| word | 90 | 91 | 92 | 93 | 94 | 95 | 96 | 97 | 98 | 99 | 0 | 1 | 2 | 3 | 4 | 5 | 6 | 7 | 8 | 9 | 10 | 11 | 12 | 13 | 14 | 15 | 16 |
|------|----|----|----|----|----|----|----|----|----|----|---|---|---|---|---|---|---|---|---|---|----|----|----|----|----|----|----|
| apple | | | sorbet | | | | | | chutney | | | | | | | | | android | | | | | | | |
| browser | | | | netscape | | | | | | | | | | | | android | | | | | | | | |
| phone | | | | telephone | | | | | | | | 911 | | | | | password | | | | | |
| european | republics | | | euro | | republics | | | euro | | hollande | | republics | | | euro | | | | |

**Case study**   In Tab. 8, we report some results from the trained word representation in the NYT dataset that ranges from 1990 to 2009. We can see that the word '*apple*' in the 90s were more about a fruit used to prepare *sorbet* or *chutney*. Since the invention of Android in the early 2000s, '*apple*' has become more related to *Android*. The same trend can be seen in the case of 'browser': it was used in personal computer (e.g., *netscape* browser) in PC era, and then became more popular in mobile devices (e.g., in *Android* system) in the mobile Internet era. As for '*phone*', it typically referred to a traditional *telephone* in 1990s, and it has become related to *911* (an universal emergency number) since 9/11 attacks. In the mobile internet era, *phone* is highly related to password since intelligent phones need password while the fixed-line *phone* does not need. These cases suggest that the proposed Word2Fun can model such semantic change to some extent.

## 6   Conclusion and Future Work

To smoothly model word meaning evolution in a single process, this paper proposes to model the change of word meaning as explicit functions. With a specific type of function like sinusoidal functions, our approach consists of a sum of many sinusoids to approximate the change of word meaning. The proposed method achieves comparable and sometimes better performance than SOTA dynamic word embedding in time-aware word clustering, temporal analogy, and semantic change detection while with fewer parameters. This work is a preliminary step to develop a sinusoidal function approximation in diachronic word representation. However, we argue this paradigm can inspire more work on a general time-series or sequential modeling research.

## Acknowledgments

This work has been supported by the Quantum Access and Retrieval Theory (QUARTZ) project, which has received funding from the European Union's Horizon 2020 research and innovation programme under the Marie Skłodowska-Curie grant agreement No. 721321.

This work has also in part been supported by MIUR (Italian Ministry of Education, University and Research) through the initiative "Departments of Excellence 2018-2022" (Law 232/2016).

We would like to thank the authors of [4] and [18] for their support in reproducing their experiments. Moreover, we would like to also thank the anonymous reviewers for the very constructive comments.

## Societal Impacts

We acknowledge that source data for training has a direct impact on learned embeddings. Documents in source data may involve various biases e.g. (e.g., political, racial, and sexual biases). Therefore, it may also introduce biases from source data to learned embedding that may affect decision making and predictions in downstream tasks. More importantly, such biases may vary over time; manipulating the ratios of documents at different time may have different results in embeddings. We encourage that future studies based on this work should consider this matter.

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
