# OpenReview forum: "Word2Fun: Modelling Words as Functions for Diachronic Word Representation"
_NeurIPS.cc/2021/Conference — NeurIPS 2021 Poster_

### Official Review · Reviewer_6eKb · 2021-07-16

**Rating:** 5
**Confidence:** 4

**Summary:**

This submission addresses the problem of learning diachronic
word representations. A new method is proposed that models
words as functions and does not rely (it is claimed) on
learning representations for discrete time intervals first
and then postprocessing them.  The new method is potentially
more powerful than prior work since it can in theory
approximate any continuous process.  It uses fewer
parameters, can model longterm processes and has improved
time complexity.  Good performance on a number of tasks is
demonstrated.


**Ethical Concerns:**

None.

**Limitations And Societal Impact:**

Yes.

**Main Review:**

Current methods for learning diachronic representations all
have some drawbacks. So it is great to see this new
proposal, which is in principle more powerful than previous
work (since it can model any continuous function) and seems
to be more sample-efficient (by not partitioning the
overall learning problem into separate small learning
problems first).

However, I have several concerns about the details of the
paper, which are discussed below. Unless these concerns are
addressed, I would lean against accepting the paper.

DISCRETE VS CONTINUOUS MODELS

I'm not sure I understood the claim that all prior work is
limited to comparing consecutive time slices through a
Markov process. Vector space alignment approaches usually
align all time slices to a single designated vector
space. This does not impose a Markovian structure on the
representations.

On lines 59 to 62, an argument is made that it doesn't
matter whether you model the process as discrete or
continuous. But Equation (2) is discrete in time. Surely,
the claim is not that how you discretize time for Equation
(2) does not matter? Could you say more about what your
discretization is? (e.g., how many time slices you use)

PRIOR WORK

There is at least one other approach that is not based on a
time-stamped corpus:

https://aclanthology.org/N18-1044/

This paper should be discussed and compared against.

TIME COMPLEXITY

The argument that O(TVD) gives you better complexity that
O(kVD) seems weak to me. T is usually relatively
small. Maybe this could be backed up by giving examples
where the number of time slices is large?

MODELING OF LONGTERM PROCESSES

"do not consider meaning evolution in longer timespan"
(lines 6/7)

It would be an interesting claim to say that there are such
longer timespan meaning evolutions that the proposed method
can model whereas prior work cannot.

However, I find this unconvincing without any analysis. It
seems to me that all prior work can model simple longer
timespan meaning evolutions, e.g., if they are simple linear
evolutions. So the question is not so much: can you model
longer timespan evolutions, but rather: do *complex* longer
timespan evolutions occur in reality? The paper lacks
analysis relevant to this question.

MODELING "POWER" OF PROPOSED METHOD

The proposed method can in principle provide a more
powerful modeling framework than prior work. However, there
is no analysis showing that it does anything interesting
over and above prior work in my opinion.

The authors do not seem to be very knowledgeable in the
linguistic details of diachronic change. A good reference to
refer to would be this:

Peter Koch. 2016. Meaning change and semantic shifts. In
Päivi Juvonen and Maria
Koptjevskaja-Tamm, editors, The lexical typology of semantic
shifts, pages 21–66. De Gruyter,
Berlin.

Koch proposes that diachronic change consists of a process
where a word with the initial meaning M1 acquires a second
meaning M2. M2 initially may have frequency close to 0, but
then increases over time. So one theory of diachronic change
is as a process of a change in frequency distribution of
underlying senses.

One could derive many interesting hypotheses from this
theory and test them using word2fun.

For example, are there any *individual* words that mix
shortterm and longterm evolutionary components in their
meaning change? If it's a simple frequency distribution
change, then maybe not.

Does an individual word show different speeds in meaning
change over time? There should be examples of that, e.g., a
very slow increase of M2 initially and then a quick fading
of M1 later.

Do different words have different time scales with respect
to meaning change? I would suspect yes, and the model (if it
actually lives up to its promise and works as intended)
should be able to show this.

Without any analysis of what the model is actually learning
(beyond some performance numbers), it is not possible for me
to determine whether it is doing something interesting
beyond prior work or not.

EVALUATION

My understanding of the semantic change evaluation is that a
non-standard setup is chosen, so it is difficult to compare
model performance to prior work. I would find it necessary
that experiments are run that guarantee direct comparability
to Semeval-2020.

MINOR POINTS

"Because of the equivalence between prediction-based word
embedding and matrix factorization[12]"

It is generally believed that the proof in [12] is not
correct, so the equivalence should not be presented as
truth, but as a conjecture.

The notion of compass (line 129) is not sufficiently
introduced and explained.







RESPONSE TO AUTHOR RESPONSE AND REVIEWER DISCUSSION


I would like to thank the reviewers for responding to my
review!

I'm not very familiar with the SemEval dataset, but what you
write makes sense. It is still unfortunate that there is no
direct comparison to prior published numbers.

> We believe that the revised paper has been improved,
> unfortunately, the submit system does not allow us to
> update the paper (as ICLR did).

This is indeed a big issue. I could imagine that I would
view the paper more favorably if I could read the updated
version, but since that's not possible, I don't feel
confident about upgrading my review.

It's a particularly big issue for this:

> Analysis of what the model is actually learning We will add
> a new section and Appendix B to quantitatively show and
> visualize what the model is actually learning.

Since I cannot look at this, I don't feel comfortable
allowing it to influence my recommendation.

> The polysemous (multiple meanings for a single word) issue
> cannot easily be solved by typical word vectors since they
> adopt a paradigm of "one vector per word", one may refer to
> "contextualized word embedding" like ELMO, BERT for this
> issue.

This is a longer discussion, but it is actually possible to
analyze single-vectors for polysemy. For example, if sense A
dominates in the first period and sense B dominates in the
last period, then the first single-vector will be close to
words with meanings similar to sense A and the last
single-vector will be close to words with meanings similar
to sense B.

I don't feel my concerns about Rosenfeld & ErK and about
analysis were addressed. I concur with my fellow reviewers
on these points:

oVhR: I would worry that if this paper were accepted they
might continue to downplay prior work in the same way,
rather than appropriately attributing the main idea of
modeling words as functions to R&E'18 as deserved.

rTwC: While, the analysis that finds that most of the words
do not complete a full period is interesting, I agree with
other reviewers that such analysis regarding several aspects
of the diachronic change is crucial to better understanding
of the effectiveness of the approach. In fact, in response
to R6eKb, the authors' proposed metric of correlation b/w
temporal change via analyzing gradients (not in the paper)
exhibits low correlation values. While this doesn't
necessarily mean that the proposed approach doesn't work,
this certainly underscores the importance of a more thorough
analysis to understand the behavior of the proposed
approach.




**Time Spent Reviewing:**

4

---

> ### Author Response · Authors · 2021-08-10
> **We modified inacurate claims, added related work, and clarified the time complexity**
>
> [**DISCRETE VS CONTINUOUS MODELS**]
>
> Thanks for your comments on the  Markovian structure. We agree that in the submitted version of the paper the discussion of this aspect is inaccurate and needs to be improved.
> We wanted to highlight that the meaning evolution is a mixture of short and long term changes and therefore cannot be reduced to one-hop changes, which are at the basis of a Markov process.
>
> Some of the previous works relies of ``one-hop change'', e.g., Bamler \& Mandt ICML 2017,  Hamilton et. al ACL 2016,  Kim et.al ACL 2014, Kulkarni et.al WWW 2015, Rudolph \& Blei WWW 2018, and Yao et.al WSDM 2018. The basic rationale of these works (including vector space alignment approaches) is to align word vectors in all time slices.
>
> In Bamler \& Mandt ICML 2017, the basic inductive bias to learn dynamic embedding is to assume the probability that transform word embedding from $t$ to $t+1$ as below (see Eq. 4 in Bamler \& Mandt ICML 2017):
> \begin{equation}
>     p(U_{\cdot,t+1} \vert U_{\cdot,t}) \propto \mathcal{N}(  U_{\cdot,t}, \delta_t^2 ) \mathcal{N}(  0, \delta_0^2 )
> \end{equation}
> while $\mathcal{N}$ is a   Gaussian distribution. The former prior aims to make two consecutive time-specific word embedding being close; its variance $\delta_t^2$  is related to the time difference between $t$-th and $t+1$-th timestamps. While the latter prior aims to  prevents the embedding vectors from growing very large. Note that $p(U_{\cdot,t+1} \vert U_{\cdot,t})$ is independent to either $p(U_{\cdot,t+2} \vert U_{\cdot,t+1})$  or  $p(U_{\cdot,t} \vert U_{\cdot,t-1})$.
>
> In Hamilton et al. ACL 2016, the goal is to find an orthogonal Procrustes $R^{t}$ to align two consecutive learned time-specific embeddings as below (see Eq. 4 in Hamilton et al. ACL 2016):
> \begin{equation}
>     R^{t} =  \underset{Q^TQ = I}{\textrm{argmin}} || U_{\cdot,t}Q - U_{\cdot,t+1} \vert \vert_F
> \end{equation}
> Note that $\{R^{t}\}_{t=1}^{T}$ are independent.
>
> Yao et al. WSDM 2018 proposed to learn dynamic word bedding by minimizing (see Eq. 5 in Yao et al.):
>
> \begin{equation}
>     \underset{ U_{\cdot,1}, U_{\cdot,2}, \cdots, U_{\cdot,t}  }{\textrm{minimizing}} \frac{1}{2} \sum^T_{t=1} \vert \vert Y(t) -U_{\cdot,t} U_{\cdot,t} ^T \vert \vert_F ^2 + \frac{\lambda}{2} \sum_{t=1}^{T} \vert \vert U_{\cdot,t} \vert \vert_F ^2 + \frac{\tau}{2}  \sum_{t=1}^{T} \vert \vert U_{\cdot,t} - U_{\cdot,t-1}  \vert \vert _F ^2
> \end{equation}
>
> $Y(t)$ is the PPMI matrix in time $t$. The only term to connect time-specific word vectors is the last one, namely, $ \frac{\tau}{2}  \sum_{t=1}^{T} \vert \vert U_{\cdot,t} - U_{\cdot,t-1}  \vert \vert _F ^2$ which  also considers only two consecutive time-specific word embedding.
>
> Other works do not rely on a one-hop change: this is the case of Di Carlo et al. AAAI 2019 which aligns all time-specific word vectors with a single compass, and Dubossarsky et al. ACL 2019, and Rosenfeld et al.
>
> In principle, our models (Word2Fun) allow one to define time $t$ as a real number since the evolution is modeled as continuous functions and any real-valued variable is valid.
> In detail, in any training triplet with $(u,v,t)$, $t$ could be any real number.
> For example, one can get the word vectors in the 10000.1 second in 2021. However, in practice, for simplicity, existing works tend to define time as integer numbers, by splitting a time span (1990-2016) as some time slices (1990, 1991,..., 2016) and the index of slices would be an identifier; these identifiers usually are integers in order, namely (1, 2,..., 17).
>
> To fairly compare with existing works, we also follow the practice that splits corpora yearly or every decade. If any corpora provide a more detailed date (like the exact month, day, and time), this model could in principle make use of this fine-grained granularity to time.
>
> We use 20 time slices in the COHA  dataset (for every decade) and 27 time slices (for each year) for the NYT dataset, as shown in Table 2.
>
> [**PRIOR WORK**]
>
> We thank the reviewer for pointing this out. Rosenfeld et al. NAACL proposed to represent time as a  word-independent variable in neural networks. Their approach is similar to our Time2Fun, a particular instance of Word2Fun. However, our approach is different from Rosenfeld et al. since we did not consider time as an independent component, later integrated through an additional component. In the revised version of the paper, we will report these remarks.
>
>
> [**TIME COMPLEXITY**]
>
> We uses 20  time slices in the COHA  dataset and 27 time slices for the NYT dataset, as shown in Table 2. If we split COHA  dataset yearly to capture semantic evolution at a fine-grained granularity, one could get 200 time slices, which will cost a lot of space for existing work.
>
> Longitudinal studies carried out by expert users, e.g., sociologists studying the discourse on public issues in the media, may require the study of the semantic evolution using splits based on quarters or months. We will mention this in the revised version.
>
>
> [**MODELING OF LONGTERM PROCESSES**]
>
> Thank you to point this out. We agree with you that the statement "do not consider meaning evolution in longer timespan'' is incorrect and unconvincing.  We will revise it properly in the next version.
>
> The  existing work did consider meaning evolution in a longer timespan, but such a process in most existing work is not treated as a unified continuous process, as expected. For example, semantic shifts are usually considered as slow and \textbf{regular} changes (Kutuzov et.al COLING 2018).
>
> This could be further explained as below.
> Let us  take linear projection (Hamilton et.al ACL 2016 and Kulkarni et.al 2015)  as an example of word meaning evolution in existing work, the transformation from word embedding in $t$ (denoted as  $U_{\cdot,t} \in \mathbb{R}^D$) to word embedding in $t+1$ (denoted as  $U_{\cdot,t+1} \in \mathbb{R}^D$) is a linear projection, namely, $U_{\cdot,t+1} = U_{\cdot,t} W^{t}$ where $w \in \mathbb{R}^{D\times D}$. Now, for a $T$-time evolution, probably a longer-term process as said, would be,
> $$
> U_{\cdot,T}  =  U_{\cdot,0} W^{0} W^{1} \cdots W^{T-1}
> $$
> Such a $T$-time evolution is represented as a product of multiple matrices ($W^{0} W^{1} \cdots W^{T-1}$). Most importantly, these matrices are independent of each other. Therefore, such a  longer-term evolution may not be smooth enough and does not form a unified continuous process.
>
>
>
>
> Regarding your question `do complex longer timespan evolutions occur in reality', we will clearly clarify in Appendix B that the long-term timespan evolution exists.  Semantic shifts are naturally separated into two important classes (Kutuzov et.al COLING 2018): linguistic drifts (slow and regular changes in core meaning of words) and cultural shifts (culturally determined changes in associations of a given word). The former usually is considered as long-term evolution that spans decades or even centuries,  while the latter is not. The used corpora COHA in this work covers 200 years which is commonly used for linguistic drifts.
>
>
> In Appendix B, we will intuitively show some case studies (visualization of learned functions). There exist some long-period sinusoidal functions after training, such that the proposed Word2Fun has the potential to capture long-term evolution by using these bigger-periods sinusoidal functions.

---

> ### Author Response · Authors · 2021-08-10
> **We conduct more study to better understand the proposed models including quantitative experiments and visulization**
>
> [**MODELING ``POWER'' OF PROPOSED METHOD**]
>
> **Diachronic change in a sense/meaning level** We thank the reviewer to provide the related reference, and we will add the above theory of diachronic change in the first paragraph of the introduction. As most works about (dynamic) word embedding (including word2vec and Glove, and also various variants of dynamic word embeddings) did, this work does not deal with diachronic meaning change in sense/meaning level. The polysemous (multiple meanings for a single word) issue cannot easily be solved by typical word vectors since they adopt a paradigm of "one vector per word", one may refer to "contextualized word embedding" like ELMO, BERT for this issue.
>
> **Mixture of shortterm and longterm evolution**  In Appendix B, we will visualize some learned functions. After training, for some words,  we can obverse from the visualization of learned functions that it learns sinusoidal functions with both big frequencies (small periods) and small frequencies (big periods), which could capture shortterm and longterm evolutionary components respectively.
>
> **Meaning change speed**
>
> The idea of investigating speed change is very interesting. We haven't carried out analyses on this aspect yet. However, we report some results we obtained from the analysis of the degree of semantic shift that can be investigated using the human annotations provided by SemEval 2020  Unsupervised Lexical Semantic Change Detection Subtask 2. The basic rationale for using the degree of change is the following: if in a given time interval (100 years in our case) the degree of change is higher, then the change speed is going to be higher; basically, we computed the ``average speed'' as the ratio of the total distance covered (the degree of change) and the total time taken (the time interval in our corpus). Meaning change speed of semantically-shifted words should be faster than semantically non-shifted words since the speed of the latter should be negligible. We will provide a new subsection in Sec. 5 to show how the learned parameters could be an indicator of meaning change speed.
> This is quantitatively evaluated by correlations with human annotations from SemEval 2020 Unsupervised Lexical Semantic Change Detection Subtask 2.
>
> | Learned parameters                          | Definition                            | Pearson | Spearman |
> |------------------|---------------------------------------|---------|---------|
> | frequencies term | $\textrm{AVG}(\vert \\mathbf{\Omega_i} \vert \) $ |  0.0958 | 0.1748  |
> | amplitude term   | $ \textrm{AVG}(\vert \\mathbf{R_i} \vert \) $      | 0.1213  | 0.0474  |
> |  bias term       | $ \textrm{AVG}(\vert \mathbf{B_i} \vert \) $      | -0.3640 | -0.2636 |
> |   amplitude/bias terms |  $ \textrm{AVG}(\vert \mathbf{R_i} \vert ) / \textrm{AVG}(\vert \mathbf{B_i} \vert )  $    | 0.4141  | 0.2358  |
>
> To show how these learned parameters reflect the meaning change speed, we take the best-performed Word2Fun as an example, namely Word2Fun III, a word $w_i$ is represented as  $\mathbf{B_i} +  \mathbf{R_i} [ \sin (\mathbf{\Omega_i^{(1)}} t);\cos (\mathbf{\Omega_i^{(2)}} t)  ]$ where  $\mathbf{B_i}, \mathbf{R_i}, \mathbf{\Omega_i} \in \mathbb{R}^D$. There are three cases that such a sinusoidal function degrade to a constant function: a) $\mathbf{B_i} = {\infty} $;  b) $\mathbf{R_i} = \textbf{0}$; or c) $ \mathbf{\Omega_i}= \textbf{0}$. Intuitively, smaller $\mathbf{B_i}$, bigger  $\mathbf{R_i}$ or $\mathbf{\Omega_i}$ indicate that the word $w_i$ is sensitive to time. In other words,\textit{ the meaning of words more likely changes over time in the case of smaller $\mathbf{B_i}$, bigger  $\mathbf{R_i}$ or $\mathbf{\Omega_i}$}.
>
> To examine the above intuition, we define the average of the absolute values for these parameters (i.e., $\textrm{AVG}(\mathbf{B_i})$, $\textrm{AVG}(\mathbf{R_i})$, and $\textrm{AVG}(\mathbf{\Omega_i}$), the average $\textrm{AVG}$ is computed over all the dimensions of vectors)  as the indicator of semantically-shift degree of a word $w_i$. Since in Word2Fun III, $\mathbf{B_i}$ is the time-unrelated term while $ \mathbf{R_i} [ \sin (\mathbf{\Omega_i^{(1)}} t);\cos (\mathbf{\Omega_i^{(2)}} t)  ]$ is the time related term. We also take the ratio between $\textrm{AVG}(|\mathbf{R_i} |)$ and $\textrm{AVG}(|\mathbf{B_i} |)$ as an indicator (in the last row in the table)
> The table shows that the empirical evaluation confirms our intuition. Especially, the Pearson correlations of the last two rows, especially $\textrm{AVG}(|\mathbf{R_i} |) / \textrm{AVG}(|\mathbf{B_i} |)$, are significant with $p< 0.05$.
> However, the result is not as good as the results in Table 6.
> The result in Table is surprising since these indicators do not consider the two specific evaluation time spans and they therefore capture the word evolution speed over the whole time span. Visualization of learned functions will be reported in Appendix B.
>
>
> We also try to use the first-order derivative of learned functions to measure the word meaning change speed. The first-order derivative for the dimensions with sine functions  is calculated as
> $$
> f'(u,t) =
>  \mathbf{R_i} \mathbf{\Omega_i^{(1)}} \cos(\mathbf{\Omega_i^{(1)}} t)
> $$
> and The first-order derivative for the dimensions with cosine functions  is calculated as
> $$
> f'(u,t) = - \mathbf{R_i} \mathbf{\Omega_i^{(2)}}  \sin(\mathbf{\Omega_i^{(2)}} t)
> $$
>
> The final indicator is designed as an average of the first-order derivative in a time span $[a,b]$, as Rosenfeld & Erk did. We use  $p$-norm to get positive numbers
> $$
>  \textrm{derivative}(a,b) = \textrm {AVG} (\sum_{t= a} ^{t=b}  \vert f'(u,t) \vert_p )
> $$
> We only consider the discrete timestamp like in (1810s, 1820s, ..., 2000s). The result is reported as below:
>
> | derivative  |  p | timespan (a,b)  | Pearson  | Spearman  |
> |---|---|---|---|---|
> | $\textrm{derivative} $  | 1  | 1810s - 2010s  |  0.2175  |  0.1174 |
> | $\textrm{derivative} $  | 1  | 1860s - 1960s  |  0.1978 |   0.1043|
> | $\textrm{derivative} $  | 2  | 1810s - 2010s  |  0.1293 |   0.0908|
> | $\textrm{derivative} $  | 2  | 1860s - 1960s  |  0.1318 |   0.1104|
>
> The performance is not as good as the first table. We suspect that this is because the bias terms $ \mathbf{B_i}$ was not considered in the  first-order derivative, while the bias terms are highly related to the semantic-shifted degrees.
>
> **Analysis of what the model is actually learning** We will add a new section and Appendix B to quantitatively show and visualize what the model is actually learning.
>
>
> [**EVALUATION**]
>
> As you mentioned, semantic change evaluation is indeed not a standard setup.
> However, the standard setting does not fit the selected baselines [6,9,14,16,26] and the proposed model.
> In detail, the standard setting only provides two groups of text: one is in the range of 1810–1860, the other is in the range of  1960–2010. There is missing time-specific text between the two time spans, namely 1860-1960; this hinders baselines from alignments.
> Indeed, the alignments of the baselines including Yao et al. WSDM 2018, Hamilton et al. ACL 2016, and  Kulkarni, WWW 2015 need an anchor to force two consecutive time-specific embedding to be close. However, the aligned anchor is missing due to the time gap/jump between 1860-1960.
> Also, the original corpora in SemEval do not provide the exact timestamp for all documents. This means we could have only two data points to learn functions. To be rigorous, we also train our models in the standard corpora provided by SemEval. Word2Fun models (in Word2Fun III setting)  do not converge at all, and they achieved nearly zero correlations.
>
> We will clarify this in the revised paper.
>
>  [**MINOR POINTS**]
>
> Thanks for pointing this out. We will revise related statements regarding the equivalence between prediction-based word embedding and matrix factorization.
>
> We will add some explanation for compass in the revised version. The compass is used as an anchor to project time-specific word embedding in different times to unified designated vector space, first introduced from Di Carlo et al. AAAI 2019.

---

> ### Author Response · Authors · 2021-08-24
> **Let us know if any further clarfications or dicussions are needed**
>
> We thank the reviewer for your detailed reviews and many useful references and suggested points, which did help us improve the paper a lot and make many vital statements more precise. The submitted version did have some wording issues since we are somehow new to this field. We believe that the revised paper has been improved, unfortunately, the submit system does not allow us to update the paper (as ICLR did).
>
> We are wondering if your concerns are still unsolved, and any discussions or suggestions are welcomed.
>
> Best,
> The authors

---

> ### Author Response · Authors · 2021-08-30
> **Thanks for your comments again -- second-round reponses**
>
> Thanks for your detailed comments again. Fortunately, we still have some time and chance for further discussions.
>
>
> **It is still unfortunate that there is no direct comparison to prior published numbers.**
>
> We assume that you are referring to R&E. In the response below, we will introduce the difference between R&E and Word2fun. Unfortunately, there are no overlapped tasks between them,  so the experimental comparison may be difficult. Anyway, we will try our best to explain the difference in a principled way.
>
> **This is indeed a big issue. I could imagine that I would view the paper more favorably if I could read the updated version, but since that's not possible, I don't feel confident about upgrading my review.**
>
> Although the paper was no updated. We already tried our best to copy the main content to the response. Let me know if any concern you addressed was not discussed.
>
>
> **Since I cannot look at this, I don't feel comfortable allowing it to influence my recommendation.**
>
>  Quantitative experiments were included in the first-round response. However,  the visualization part of the learned functions is invisible. We tried to explain the most important information conveyed by the visualization, that is, "70%  functions do not complete a whole period" from the visualization.
>
>
> **This is a longer discussion, but it is actually possible to analyze single-vectors for polysemy. For example, if sense A dominates in the first period and sense B dominates in the last period, then the first single-vector will be close to words with meanings similar to sense A and the last single-vector will be close to words with meanings similar to sense B.**
>
> We could definitely do it. I also noticed that R&E did this. The main concern from us is that dynamic word embedding which adopts the **one vector per word paradigm** is ineffective in this matter (modeling word meaning at sense level); since there is not direct inductive bias for sense-level evolution from training data itself. We could add a case study (a few examples) to show that it may capture sense-aware evolution to some extent. However, it is not easy to find quantitative metrics to check to which extent that dynamic word embedding related approaches could capture sense-aware evolution. Even in R&E, no quantitative metrics were used. We feel that it is unconvincing to conduct such a case study to show Word2fun could slightly capture sense-aware evolution.
>
> To deal with the polysemy issue at the sense level, we believe that word vectors-based approaches (like the proposed Word2Fun) that assume **one vector per word** are not the fundamental solution. One may have to inject sense-involved knowledge (e.g., WordNet) into word representation, like SenseBERT (https://arxiv.org/abs/1908.05646) and Hu et.al (https://aclanthology.org/P19-1379.pdf).
>
> Nevertheless, we would like to include the case study in the appendix of the revised version if it is accepted.
>
>
> **oVhR: I would worry that if this paper were accepted they might continue to downplay prior work in the same way, rather than appropriately attributing the main idea of modeling words as functions to R&E'18 as deserved.**
>
> As suggested,  we acknowledge and accept that the main idea of modeling words as functions  (in our introduction) should be attributed to R&E. We promise that we will do it. This promise would be guaranteed because reviews would be public (we opt to make it public no matter it will be accepted or not), and one may risk promising to do something but not to do during openreview rebuttal.
> We believe that we could solve the above concerns by accurately attributing credits to the right papers via wording, especially in the openreview scenario.
>
> From a mathematical point of view, any neural network could be a function to map the input to output. Neural networks that are trained by triples skip-grams (u,v,t) using noise contrastive estimation (including earlier or later dynamic word embedding related work) are in principle "functions of words", that is, $ NueralNetwork(w,t) \in \mathbb{R}^D $ that maps word $w$ in time $t$ to a vector.
> R&E is the first one that **uses time as a continuous variable**. Similar motivations could be found in Time2vec (https://arxiv.org/abs/1907.05321) and more generally position embedding Vaswani et.al (https://arxiv.org/abs/1706.03762) and Wang et.al (https://openreview.net/forum?id=Hke-WTVtwr).
>
> We also want to highlight the difference between Time2fun and Word2fun in our paper -- the former (Time2Fun) is a special case of the latter (Word2Fun) in this paper. The difference refers to whether time-aware word representation is decomposable. Time/word encoders of Time2fun are decomposable, and time/word encoders of Word2fun are indecomposable, defined here.
> "Can the time-aware representations be decomposed by two separate encoders $f_{time}$, f_{word} and then combine them as final encoder using arbitrary operations?" Based on this, we could conclude that  R&E should more likely go to the time2fun category (not word2fun) since it is decomposable as stated below. Basically, it is also safe to state that  $$u_{it} = f_{word}(i) * f_{time}(t) $$ are word functions over time, in a specific case that the  time encoder and time encoder are decomposable.
>
> Although R&E does not clearly mention dynamic/temporal/diachronic word embedding, we could easily know that R&E do acuautlly imply a word embedding and time embedding by decomposing the neural networks,
> $$f_{word} (w) =  T\vec{w} +B$$
> and
> $$f_{time} (t) = tanh(M_2 (M_1 t + b_1)  + b_2)$$
>
> And a word in time t is represented
> $$U_{it} = f_{word}(w) * f_{time}(t) $$
> $*$ is a matrix-vector product.
> This is like our time2fun case:  $U_{wt} = f_{word}(w) + g_{time}(t)$ if we do not consider the last output layer on the top of $U_{it} $.  We agree that R&E has a more effective interaction module than the proposed Time2Fun (the simple case of word2fun) thanks to the matrix-vector product and final output layer.
>
> Overall, we will introduce the four fundamental differences between R&E and Word2fun:
> > - The time/word encoders in R&E are *decomposable* since time encoder and word encoders could be decoupled like Time2Fun.  While  Word2Fun  learns word functions in which time and word encoders are indecomposable. Decomposable encoders may bring some drawbacks, as stated by the first response to reviewer dgc4.
> > - The operations of R&E are time-consuming. Because matrix-vector product in R&E is more expensive than addition, and R&E involves a three-order tensor in word encoder, and two non-linear layers for time encoder.
> > - We use non-symmetrical encoders of input encoder and output encoder, one is temporal, one is static; the latter works as a compass to align word embedding. R&E, which uses temporal encoders for both input/target encoders;  both non-static input/target encoders may suffer from the rotation invariant issue as stated in the Sec. 2.2 of the paper.
> > - We have a theoretical guarantee (although as an upper bound) that the proposed Word2fun could approximate any word meaning evolution. Any design of Word2fun could find concrete motivations behind it: Word2fun uses a standard skip-gram model with negative sampling, uses continuous functions for temporal word embedding, uses sinusoidal functions for better approximations,  uses a compass for further alignment as Di Carlo do. While there is no theoretical upper bound for R&E to approximate any word meaning evolution, and many designs seems intuitive -- it may be reasonable but the motivations were not well-discussed
>
>
> **rTwC: In fact, in response to R6eKb, the authors' proposed metric of correlation b/w temporal change via analyzing gradients (not in the paper) exhibits low correlation values. While this doesn't necessarily mean that the proposed approach doesn't work, this certainly underscores the importance of a more thorough analysis to understand the behavior of the proposed approach.**
>
> As we explained in the previous response to 6eKb (the last paragraph for that comment), the bias terms were not included in the first-order derivative, while the bias terms are highly related to the semantic-shifted degrees (see the absolute numbers of the last second row in the first table). In a simple case, e.g., $f1(u,t) = sin(\omega x) + 0.00001$ and  $f2(u,t) = sin(\omega x) + 10000$. Their first-order derivatives of $f1$ and $f2$ are the same (namely $f1'(u,t)= f2'(u,t) = cos(\omega x) $), but the latter $f2$ is nearly a constant function (ranging in [9999,10001]) which suggests that its word meaning may not change too much over time (especially when we measure between-word relatedness using cosine similarity in a sense only angle matters),  the former $f1$ should change over time (ranging in [-0.99999,1.00001]). Therefore, bias terms matter in word meaning change speed (see the first table) while it is not considered in first-order derivatives.  We hope this explanation could help a little bit.

---

> ### Author Response · Authors · 2021-09-02
> **Case study as required from reviewer 6eKb -- analyze single-vectors for polysemy**
>
>
>
> The following table reports on a case study to show that the proposed Word2Fun could at least to some extent capture word meaning change regarding words with multiple senses. We used the word **gay** for the case study.
>
>  |word | 1900s|1920s|1940s|1960s|1980s|2000s  |
> |-------|--------------|---------|---------| ---- | ---|---|
> |frolicsome|**0.5230**|0.3574|0.2802|0.1511|0.1649|0.1992|
> |playful|0.4094|0.3757|**0.4268**|0.3298|0.2425|0.2839|
> |debonair|0.3840|0.4705|**0.5523**|0.4597|0.2243|0.3547|
> |activists|0.2319|0.2430|0.0892|0.2894|**0.4698**|0.4072|
> |homosexuality|-0.1435|-0.0274|0.1209|0.2605|0.3242|**0.3727**|
>
> The table reports words similar to **gay** by using the representation obtained through Word2Fun III and the following measure of similarity:
>
> $$sim(\vec{w}_1, \vec{w}_2 ) = \frac{ \vec{w}_1  \vec{w}_2 ^ T} { \vert \vec{w_1}  \vert \cdot \vert \vec{w}_2 \vert}$$
>
> From the table, we could see that the sense of **playful** decreases over time, while the sense of **homosexuality** increases.

---

### Official Review · Reviewer_oVhR · 2021-07-16

**Rating:** 6
**Confidence:** 4

**Summary:**


This paper presents a new approach to modeling word embedding change over time: each word $w_i$ is represented as a function $g_i(t) : N \rightarrow R^D$ that maps a timestamp $t \in \mathbb{N}$ to an embedding $e_{it} \in \mathbb{R}^D$. The function $g_i(t)$ can then be parameterized in a variety of ways; the paper focuses primarily on sinusoidal parameterizations. The paper motivates this approach via the Weierstrass theorem, from which it follows that a real function on the interval [a,b] can be well approximated by a trigonometric polynomial.

The key contribution of this paper is that it breaks out of the typical paradigm that has been used for diacronic word embeddings: i.e., that a separate word embedding $e_{it}$ is explicitly learned for each word $w_i$ and each timestamp $t$. Doing so comes with several drawbacks that this paper addresses: (1) that an alignment between timestamps must be learned (2) that the alignments are (usually) only dependent in a pairwise Markovian fashion, thus missing out on longer time effects (3) that space required is high.

This paper evaluates its method with different parametric functions (e.g. linear, word/time independent, and various sinusoids) on several benchmarks established in prior work for the evaluation of diacrhonic word embeddings and attains (mostly) better results than strong baselines from prior work.


**Ethical Concerns:**

None.

**Limitations And Societal Impact:**

No -- see the first "weakness" mentioned above, which highlights an important limitation of this work.


**Main Review:**

Strengths:
- The paper was very clearly written and packs a lot into its narrative. In particular, S2.2 and S4.2 exemplify the paper's nack for explaining dense technical ideas in a concise yet clear way.
- This paper is obvious in retrospect in the best possible way. From the title, one might think that this paper sets out to accomplish something difficult (modeling words as functions), but reveals quickly that the idea itself is extremely simple (i.e. simply parameterizing the functions in natural ways and training by backprop). The beauty of its contribution is that no one seems to have thought of doing so before.
- The experiments suggest that this method works empirically better than prior work despite using dramatically fewer parameters.

Weaknesses:
- The most significant problem with the paper is the lack of nuance in describing word change over time. For example, the initial motivating example in the Introduction is an example of a new sense of a word appearing that overtakes the previous senses of the word in popularity. Such a change is not well-captured by a smoothly varying function over time at all! In fact, it would be better captured by a model which allows the introduction of additional senses over time that (may or may not) correlate with past senses of the same word. The paper makes no attempt to explain this mismatch between motivating example and method. (Having said this, such an egregious error could be fixed in the writing. Doing so would require re-motivating this approach through the lens of words whose change over time does NOT occur do to the introduction of a new sense, but rather through the more gradual change of their use over time in relation to other words.) Why is this so important? Because the paper does not at present adequately explain *why* it might be outperforming other methods.
- The main piece that is missing from this paper is additional analysis into the types of functions that are being learned. (At present the only hint of analysis given is a very brief anecdote about how nearest neighbors of specific words vary over time.) What do the learned functions look like? Are there periodic aspects to them or do they stretch out the sinusoids in order to avoid periodic looking functions? Can we learn anything about word change by investing the shape of the functions across larger groups of words?
- The paper never mentions other possible applications of the idea behind representing words as functions. For example, instead of modeling how words vary over time, one could add another dimension (e.g. genre) and explore how words vary over that dimension.
- No attempt is made at bridging the gap between the Weierstrass theorem + corralaries, which applies to functions defined on closed intervals, and the true problem of learning word embeddings on open intervals.



**Time Spent Reviewing:**

1.5

---

> ### Author Response · Authors · 2021-08-10
> **We changed the motivating examples, showed what the models really learn by quantitative experiments and visualization, possible applications, etc.**
>
> [**motivating examples**]
>
> Thanks for your comments on the motivating examples. In the revised version, we will use another motivating example, for instance, the word *Trump* with a trajectory *real estate* $\longrightarrow$ *television* $\longrightarrow$ *republican*. The word *Trump*  is a mixture meaning of  (a) *the president of Trump's real estate*,  (b) *a media Giant for television*, and (c) *a republican who finally becomes the president of US*.
> In New York Times, *Trump* was first known as (a), and then (b). Finally his best-known role is the *republican* as (c). We will visualize this example with learned dynamic word embedding in Sec. 5.3.
>
> Another example could be the word *nuclear*; the word is related to the nucleus of an atom. However, if we consider the use of the word in the media in relation to other words, we can expect a strong relation in the 1930s and 1940s with the "weapon'' topic/theme; the relation with *weapon*' will be also strong till to the end of the cold war. At the end of the 1980s, because of the Chernobyl accident, we will expect to see a strong relationship  with "energy'' (nuclear power) and the debates on the danger related to the nuclear plants. This debate rose again after the The Fukushima Daiichi nuclear accident in 2011.
>
> [**What do the learned functions look like?**]
>
> We will add a new subsection in  Sec. 5 and Appendix B to show what the learned functions look like by using quantitative analysis and visualization of learned functions.
>
> **Are there periodic aspects to them or do they stretch out the sinusoids in order to avoid periodic looking functions**
> The analysis in COHA dataset shows that roughly 70\% of sinusoidal functions do not complete a whole period in 20 decades (from 1810s to 2010s) -- this is evaluated for the 37 test words in the SemEval task in a 50-dimensional Word2Fun III setting.
> Interestingly, a few frequencies of semantic-shifted words are very small, e.g., smaller than $1e^{-8}$, resulting in nearly constant functions (because of small frequencies and very big periods) -- which means these words do not change their meaning over time.
>
> [**Can we learn anything about word change by investing the shape of the functions across larger groups of words?**]
>
> The shape of the functions is reflected by their amplitude,  frequencies, and biases. These (including amplitude,  frequencies, and biases) could be indicators for how much the word changes over time, or called `semantic shift degree'.
>
> | Learned parameters                          | Definition                            | Pearson | Spearman |
> |------------------|---------------------------------------|---------|---------|
> | frequencies term | $\textrm{AVG}(\vert \\mathbf{\Omega_i} \vert \) $ |  0.0958 | 0.1748  |
> | amplitude term   | $ \textrm{AVG}(\vert \\mathbf{R_i} \vert \) $      | 0.1213  | 0.0474  |
> |  bias term       | $ \textrm{AVG}(\vert \mathbf{B_i} \vert \) $      | -0.3640 | -0.2636 |
> |   amplitude/bias terms |  $ \textrm{AVG}( \vert \mathbf{R_i} \vert ) / \textrm{AVG}( \vert \mathbf{B_i} \vert )  $     | 0.4141  | 0.2358  |
>
> To show how these learned parameters reflect the semantic shift degree, we take the best-performed Word2Fun as an example, namely Word2Fun III, a word $w_i$ is represented as  $\mathbf{B_i} +  \mathbf{R_i} [ \sin (\mathbf{\Omega_i^{(1)}} t);\cos (\mathbf{\Omega_i^{(2)}} t)  ]$ where  $\mathbf{B_i}, \mathbf{R_i}, \mathbf{\Omega_i} \in \mathbb{R}^D$. There are three cases that such a sinusoidal function degrade to a constant function: a) $\mathbf{B_i} = {\infty} $;  b) $\mathbf{R_i} = \textbf{0}$; or c) $ \mathbf{\Omega_i}= \textbf{0}$. Intuitively, smaller $\mathbf{B_i}$, bigger  $\mathbf{R_i}$ or $\mathbf{\Omega_i}$ indicate that the word $w_i$ is sensitive to time. In other words,\textit{ the meaning of words more likely changes over time in the case of smaller $\mathbf{B_i}$, bigger  $\mathbf{R_i}$ or $\mathbf{\Omega_i}$}.
>
> To examine the above intuition, we define the average of the absolute values for these parameters (i.e., $\textrm{AVG}(\mathbf{B_i})$, $\textrm{AVG}(\mathbf{R_i})$, and $\textrm{AVG}(\mathbf{\Omega_i}$), the average $\textrm{AVG}$ is computed over all the dimensions of vectors)  as the indicator of semantically-shift degree of a word $w_i$. Since in Word2Fun \RN{3}, $\mathbf{B_i}$ is the time-unrelated term while $ \mathbf{R_i} [ \sin (\mathbf{\Omega_i^{(1)}} t);\cos (\mathbf{\Omega_i^{(2)}} t)  ]$ is the time related term. We also take the ratio between $\textrm{AVG}(|\mathbf{R_i} |)$ and $\textrm{AVG}(|\mathbf{B_i} |)$ as an indicator (in the last row in the table)
> The table shows that the empirical evaluation confirms our intuition. Especially, the Pearson correlations of the last two rows, especially $\textrm{AVG}(|\mathbf{R_i} |) / \textrm{AVG}(|\mathbf{B_i} |)$, are significant with $p< 0.05$.
> However, the result is not as good as the results in Table 6.
> The result in the Table is surprising since these indicators do not consider the two specific evaluation time spans and they, therefore, capture the word evolution speed over the whole time span. Visualization of learned functions will be reported in Appendix B.
>
> The above table is also shown for reviewer 6eKb.
>
> [**possible applications**]
>
> Thanks for your comments. In the revised version,  we will mention that there are various applications of Word2fun. One of the most interesting applications with dynamic user profiles.
> In a typical static item recommendation scenario,  the basic goal is to approximate user-item preference scores by a dot product between a static user embedding (denoted as $\vec{u}$) and item embedding (denoted as $\vec{i}$). $ f_{u,i} \propto  \vec{u} \vec{i} ^T $, while  $p_{u,i} $ is  a scalar that indicates rating score or buying/like behaviors. When extending it to temporal process, the temporal preference  $p_{u,i}(t) $ would be sequences of scalars that indicates user-item interaction behavior over time; such that the user profiles would also evolve with time $t$, denoted as $\vec{u}(t) = f(t)$. This could be an equivalent problem as stated in this paper and  $p_{u,i}(t)$ could be approximated by a sum of sinusoids.  More interestingly, the periodical properties of sinusoidal functions could be used to model repeating purchase behavior, for example, one may buy beers every month and buy coats every winter.
>
> **add another dimension (e.g. genre) and explore how words vary over that dimension**  Treating genre as another dimension is a good idea. One concern to modeling genre as another dimension is that genre itself is not sequential like time,  therefore continuous functions over genres may not make sense.
>
> [**gap between the closed intervals in theorem and the open intervals in true problem**]
>
> Actually, in true problems, the domain is also always closed in word embedding since there always exists a maximum time stamp and a minimum time stamp.
>
> In this work, we did no claim/expect that the proposed Word2Fun  can recall the word meaning in past (before the first date of training corpora) or predict the word meaning in the future, although it seems that Word2Fun possibly could have a little potential. We will clarify this in the revised version.

---

### Official Review · Reviewer_dgc4 · 2021-07-18

**Rating:** 7
**Confidence:** 3

**Summary:**

This paper proposed a new paradigm to learn diachronic word representations, which models words as functions of time, so that the word meaning in a different time could be correlated and evolves within a continuous process. The paper adopted various functions such as linear function and sinusoidal functions to approximate the word meaning evolution in the context of distributed representations. The experimental results show that this approach is promising in the time-aware NLP tasks like time-aware word clustering, temporal analogy, and semantic change detection.

**Limitations And Societal Impact:**

Please see the main review for further details.

**Main Review:**

Generally, the topics of the paper and the idea of modeling words as functions of time are super interesting. The paper is easy to follow until section 4.1, before which all the motivations are clear and straightforward. My main concern is why not use a neural network to learn the approximation in Eq. 3? E.g., adding a time embedding to a distributed model. I suggest that a more convincing motivation behind the Polynominal approximation should be provided.

Strengths:
* The paper provided a really good background section, which helps the reader better understand the problem settings and related work.
* The experiments includes lots of time-aware tasks that validates the benefits of the methods.

Weaknesses
* Word2Fun does not improve significantly compared to Compass aligned Word2Vec in Table 4 and Table 5.
* The motivation of using polynomials is not clear.

Questions:
* Did you consider polysemous words at the same timestamp?

**Time Spent Reviewing:**

3

---

> ### Author Response · Authors · 2021-08-10
> **Response for comments from Reviewer dgc4**
>
>  [**why not add  a time embedding to a distributed model**]
>
> We did implement a baseline called `Time2Vec' (adding a time embedding to a distributed model) but we did not report the final result because it is ineffective in these downstream tasks (similar to Time2Fun). Time2Vec is a similar setting to Time2Fun of this paper. The difference between time2vec and time2fun is like the difference between fully-learnable position embedding and sinusoidal position embedding in transformers; they (fully-learnable position embedding and sinusoidal position embedding) were claimed to have nearly identical performance (Vaswani et.al NIPS 2017) while the latter could be generalized to longer spans thanks to the periodical property of sinusoidal functions.
>
> Note that Time2Fun and Time2Vec models separately learn word-free time representation and a time-free word vector representation. This may cause some issues. Let us take a simple case as an example:  word representation is a sum between a word vector and a time vector. Namely, a word $w_i$ in time $t$ is represented as
> $$
> U_{i,t} =  f_i + g_t
> $$
> Where $f$ is the word embedding and $g$ is the time embedding. Now we consider two cases
>
> **Case 1: different words in the same timestamp**
> The difference between any two words is always  constant with respect to time:
> $$
> U_{i,t} - U_{j,t} =   (f_i + g_t) -  (f_j + g_t) = f_i - f_j
> $$
> The result is time-unrelated, indicating any relationship between words does not evolve over time -- it seems that this does not make sense since we usually assume the between-word relationship should change. For example, *Trump* and *president* should not be always close in both 2018 and 2021.
>
> **Case 2: same word at different time**
>  For any word, the difference between its representation in $t_1$ and $t_2$ is constant.
>  $$
> U_{i,t_1} - U_{i,t_2} =   (f_i + g_{t_1}) -  (f_i + g_{t_2}) = g_{t_1} - g_{t_2}
> $$
> The result is word-free. This leads to a contradiction with the fact that some words may evolve with time and some of them may not.
>
> To overcome the above two issues, one has to design some much more complicated interaction modules between a word component and a time component. Such a design may have some other costs like complexity, interpretability, or efficiency.  We leave it as future work. We will clarify the above in the revised version.
>
>  [**The motivation of using polynomials**]
>
> Using polynomials is the typical way for function approximation.
> Polynomials are used to approximate *any* function defined over [a,b] in R. The theorem is a therefore fundamental result and we used it because of its generality. In other words, polynomials are to functions as rational numbers are to real numbers.
>
>  [**whether consider polysemous words at the same timestamp**]
>
> In this paper, we did not deal with the polysemous phenomenon.  This is because typical word vectors based methods cannot easily deal with the polysemous phenomenon since they adopt a paradigm of "one vector per word". The polysemous phenomenon may be handled by a contextualized version of dynamic word embedding.
>
> Considering polysemous words in Word2Fun could be an interesting extension that is worthy of exploration, while we leave it as future work.

---

### Official Review · Reviewer_rTwC · 2021-07-21

**Rating:** 7
**Confidence:** 4

**Summary:**

This paper proposes a new paradigm for modeling temporal word embeddings. Instead of the commonly prevalent approaches that train static embeddings for documents belonging to each timestamp followed by an alignment step, this work attempts to learn embeddings as a continuous function of time. This work proposes function parametrizations that use sinusoidal polynomials which can model arbitrary continuous functions. The approach is empirically compared against relevant baselines on time-aware clustering, temporal analogy, and semantic change detection.

**Limitations And Societal Impact:**

The authors claim that their work would have no societal impact. I agree to a great extent because the contribution is algorithm oriented and the paper does not touch on potential applications or deployment.

That said, since the space is provided for reflection on potential negative impact of this work, I think a discussion on the impact of source data used for learning embeddings on the correlations induced by the embeddings would be helpful in more deeply understanding the paper's underlying goal. Documents across several time periods are not all equal: they hide complex decisions around archival choices, bias in publication, etc. Acknowledging this would avoid blind spots in any future studies that use this work to test/propose social science theories.

**Main Review:**

Positives:

-- The proposed approach convincingly outperforms most of the baselines while being competitive with dynamic word embeddings. The various ablation of the proposed approach also highlight that modeling frequency parameters are more useful than explicitly modeling phase parameters.

-- The proposed approach is a natural way of thinking about temporal embeddings and is a logical succession to the traditional temporal word embeddings.

Negatives:

-- The loss function seems to work but it seems heuristics-oriented and the motivation behind the loss is weak. What is the motivation behind using both time-sensitive f(.) and compass embeddings h(.) in the loss? Does it relate to any other probabilistic objective functions?

-- In table 4, the proposed approaches trail by compass aligned word2vec by a significant margin on test1. While the prposed approach outperforms the same baselines appreciably on test 2 in table 5. Hence, the proposed explantation of compass embeddings using word2vec implementation is unsatisfactory. More ablative and qualitative analysis focusing on differences between test2 and test1 would shed more light on the differences in behavior of the proposed approach and compass aligned embeddings.

-- Analysis of learned frequencies across dimensions (qualitative or quantitative)  would help in understanding the dimensions of temporal variation in greater detail. Especially, are their dimensions with small frequencies? What do they signify because this suggests periodic oscillations.

-- Overall, the paper could benefit from deeper engagement with downstream applications or findings from the temporal embeddings. How exactly are they useful or important?

-- typos: line 166 y(t) missing the i index, line 140, algorithm 1-lines 2 and 3 overload the variable t.


**Time Spent Reviewing:**

7

---

> ### Author Response · Authors · 2021-08-10
> **We clarified the motivation of the loss function,  added qualitative or quantitative analysis analysis of learned frequencies, and introduced some possible downstream applications**
>
> [**the motivation behind the loss and its relation to probabilistic objective functions**]
>
> Skip-gram models basically learn two sets of word embedding, one is called "context embedding" and the other is called "target embedding". For each skip-gram $(u,v)$, the objective is to shorten the distance between $u$ in context embedding and $v$ in the target embedding by maximizing their dot product. Therefore, we have $f(\cdot)$ as the context embedding and $h(\cdot)$ as the target.
>
>
> The  $f(\cdot)$  is used for temporal word embedding and it is therefore time-sensitive.
> The static compass  $h(\cdot)$ is the way to align the time-specific word embedding in different times. Without the compass, time-specific word embedding may be rotated with angles,  the Alignment Issues is stated in Sec.  2.2 since word embedding is arbitrarily Rotation-invariant.
>
> One may use the context embedding $f$ as the static one and $h$ as the temporal one, this is also reasonable since skip-gram is a symmetrical relationship between words. Due to the fact that there is no notable difference to choose which one being temporal/static, we follow Di Carlo et al. to make context embedding $f$ being temporal and the target embedding $h$ being static.
>
>
>
> For the  objective function, the loss function is a typical \textit{cross entropy loss}. Suppose that we have a word pair $u,v$, its predicted probability to be a skip-gram is $ \textrm{sigmoid} (f(u,t) h(v)^T) $; its probability not to be skip-gram is
> $$
> 1 -  \textrm{sigmoid} (f(u,t), h(v)^T )=  \textrm{sigmoid} ( - f(u,t) h(v)^T)
> $$
> This is because that $1 -  \textrm{sigmoid}(x) =  1 -\frac{1}{ 1+ e^{-x}} = \frac{1+ e^{-x} -1}{1+ e^{-x}} = \frac{ e^{-x} }{1+ e^{-x}} = \frac{ 1 }{1+ e^{x}} = \textrm{sigmoid}(-x)$.
> Therefore, the predicted probability distribution is
> $$
>  p_\textrm{predicted} =
> \begin{pmatrix}
> \textrm{sigmoid} (f(u,t), h(v)^T),  \\
> \textrm{sigmoid} (-f(u,t), h(v)^T)  \\
> \end{pmatrix}
> $$
> while its ground truth probability distribution is
> $$
>  p_\textrm{pos} =
> \begin{pmatrix}
> 1  \\
> 0  \\
> \end{pmatrix}
> $$ when $u,v$ is the correct skip-gram, or
> $$
>  p_\textrm{neg}=
> \begin{pmatrix}
> 0  \\
> 1  \\
> \end{pmatrix}
> $$
> when  $v$ is the negatively sampled word as negative example.
>
> One has to minimize the difference between the predicted probability distribution and the ground truth probability distribution.   A typical way is to use the cross entropy loss for positive examples as below:
> $$
> \textrm{XEntropy} (  p_\textrm{predicted},p_\textrm{pos})=- \log \left( \textrm{sigmoid} (f(u,t), h(v)^T) \right)
> $$
>
> For negative examples,
> $$
> \textrm{XEntropy} (  p_\textrm{predicted},p_\textrm{neg}) =- \log \left( \textrm{sigmoid} (-f(u,t), h(v)^T)  \right)
> $$
>
> Since the numbers of positive and negative examples are imbalanced: for each skip-gram pair, we have to sample $\vert \hat{\mathbb{V}} \vert$ negative samples. To balance the positive examples and negative examples, we reweight training examples by giving a weight of $1$ for the loss of positive examples and giving a weight of $\frac{1}{ \vert \hat{\mathbb{V}} \vert}$ for negative examples. This results in the used loss function in the paper:
> $$
>    L =- \sum_{ (u,v, \hat{\mathbb{V}},t) \in \mathbb{D}_\textrm{train} } \left( \log (\delta(f (u,t) h(v) ^T ))  +  \frac{1}{  \vert  \hat{\mathbb{V}} \vert }  \sum _{\hat{v}_i \in   \hat{\mathbb{V}}}  \log (\delta(-f (u,t) h(\hat{v}_i) ^T ))  \right)
> $$
>
> Thanks for pointing this out, we will clarify this in the revised paper.
>
> [**qualitative or quantitative analysis in detail**]
>
> We will add a new subsection in Sec. 5 and Appendix B to show what the learned functions look like both in qualitative and quantitative ways, including histograms for parameter distributions. The analysis shows that roughly 70\% of sinusoidal functions do not complete a whole period in 20 decades (from 1810s to 2010s).
>
> For quantitative analysis,  we test the interpretable physical meaning for learned parameters by using annotated data in SemEval 2020 Un-supervised Lexical Semantic Change Detection subtask 2.  These parameters  (including amplitude,  frequencies, and biases) could be indicators for how much the word changes over time, or called "semantic shift degree".
>
> | Learned parameters                          | Definition                            | Pearson | Spearman |
> |------------------|---------------------------------------|---------|---------|
> | frequencies term | $\textrm{AVG}(\vert \\mathbf{\Omega_i} \vert \) $ |  0.0958 | 0.1748  |
> | amplitude term   | $ \textrm{AVG}(\vert \\mathbf{R_i} \vert \) $      | 0.1213  | 0.0474  |
> |  bias term       | $ \textrm{AVG}(\vert \mathbf{B_i} \vert \) $      | -0.3640 | -0.2636 |
> |   amplitude/bias terms |  $ \textrm{AVG}( \vert \mathbf{R_i} \vert ) / \textrm{AVG}( \vert \mathbf{B_i} \vert )  $     | 0.4141  | 0.2358  |
>
> To show how these learned parameters reflect the semantic shift degree, we take the best-performed Word2Fun as an example, namely Word2Fun III, a word $w_i$ is represented as  $\mathbf{B_i} +  \mathbf{R_i} [ \sin (\mathbf{\Omega_i^{(1)}} t);\cos (\mathbf{\Omega_i^{(2)}} t)  ]$ where  $\mathbf{B_i}, \mathbf{R_i}, \mathbf{\Omega_i} \in \mathbb{R}^D$. There are three cases that such a sinusoidal function degrade to a constant function: a) $\mathbf{B_i} = {\infty} $;  b) $\mathbf{R_i} = \textbf{0}$; or c) $ \mathbf{\Omega_i}= \textbf{0}$. Intuitively, smaller $\mathbf{B_i}$, bigger  $\mathbf{R_i}$ or $\mathbf{\Omega_i}$ indicate that the word $w_i$ is sensitive to time. In other words,\textit{ the meaning of words more likely changes over time in the case of smaller $\mathbf{B_i}$, bigger  $\mathbf{R_i}$ or $\mathbf{\Omega_i}$}.
>
> To examine the above intuition, we define the average of the absolute values for these parameters (i.e., $\textrm{AVG}(\mathbf{B_i})$, $\textrm{AVG}(\mathbf{R_i})$, and $\textrm{AVG}(\mathbf{\Omega_i}$), the average $\textrm{AVG}$ is computed over all the dimensions of vectors)  as the indicator of semantically-shift degree of a word $w_i$. Since in Word2Fun \RN{3}, $\mathbf{B_i}$ is the time-unrelated term while $ \mathbf{R_i} [ \sin (\mathbf{\Omega_i^{(1)}} t);\cos (\mathbf{\Omega_i^{(2)}} t)  ]$ is the time related term. We also take the ratio between $\textrm{AVG}(|\mathbf{R_i} |)$ and $\textrm{AVG}(|\mathbf{B_i} |)$ as an indicator (in the last row in the table)
> The table shows that the empirical evaluation confirms our intuition. Especially, the Pearson correlations of the last two rows, especially $\textrm{AVG}(|\mathbf{R_i} |) / \textrm{AVG}(|\mathbf{B_i} |)$, are significant with $p< 0.05$.
> However, the result is not as good as the results in Table 6.
> The result in the Table is surprising since these indicators do not consider the two specific evaluation time spans and they, therefore, capture the word evolution speed over the whole time span. Visualization of learned functions will be reported in Appendix B.
>
>
> **are their dimensions with small frequencies**
> a few frequencies of semantic-unshifted words (annotated by SemEval 2020) are very small, e.g., smaller than $1e^{-8}$, resulting in nearly constant functions (because of small frequencies and very big periods) -- which means these words do not change their meaning over time.
>
> [**downstream applications and why is it important**]
>
> There are many downstream applications of temporal embeddings. They are not only beneficial to linguistics to better analyze and understand languages, but also for journalists or sociologists who often carry out longitudinal studies to investigate how some themes/topics/issues are discussed in the media and/or perceived by the public. Some existing works, e.g.,
>
>  Nikhil Garg, Londa Schiebinger, Dan Jurafsky, and James Zou. Word embeddings quantify 100 years of gender and ethnic stereotypes. PNAS. 2017.
>
> could also explore temporal embeddings to quantify historical trends for gender and ethnic stereotypes/biases during centuries.
>
> Moreover, using sinusoids for sequential modeling could be general for time-series tasks.
> In the revised version,  we will mention that there are various applications of Word2fun. A possible application is dynamic user profiles.
> In a typical static item recommendation scenario,  the basic goal is to approximate user-item preference scores by a dot product between a static user embedding (denoted as $\vec{u}$) and item embedding (denoted as $\vec{i}$). $ f_{u,i} \propto  \vec{u} \vec{i} ^T $, while  $p_{u,i} $ is a scalar that indicates rating score or buying/like behaviors. When extending it to temporal process, the temporal preference  $p_{u,i}(t) $ would be sequences of scalars that indicates user-item interaction behavior over time; such that the user profile would also evolve with time $t$, denoted as $\vec{u}(t) = f(t)$; the approach is similar to that adopted in this paper since the time-specific user-item preference (like a skip-gram model) is also formalized as a dot product between a temporal embedding and a static embedding; such a temporal embedding $p_{u,i}(t)$ could be approximated by a sum of sinusoids.
> More interestingly, the periodical properties of sinusoidal functions could be used to model repeating purchase behavior, for example, one may buy beers every month and buy coats every winter.
>
> [**typos**]
> We will revise them and thank you to point them out.
>
> [**Societal Impact**]
>
> We will discuss these societal Impacts as suggested.

---

> > ### Comment · Reviewer_rTwC · 2021-08-25
> > **Thanks for your response**
> >
> > Thank you for responding to my review!

---

### Author Response · Authors · 2021-09-01
**Experimental comparison and difference between Rosenberg \& Erk's work and the proposed  Word2Fun.**


We would like to thank all the reviewers for the useful criticisms and suggestions.

We would also like to confirm the acknowledgment of Rosenberg and Erk (R\&E)'s work which precedes ours regarding the view of words as functions.  As said, the open nature of the review system will ensure all of us that the paper will reflect the discussion of the rebuttal phase. That said, we fully sympathize with the reviewers for the missing citation which was overlooked by us in the first version of the paper. We apologize for that.

In addition, we retrieved the code of R\&E's paper thanks to private communication with them. We re-implemented the code to make it consistent with ours and then replicated the experiments to compare Word2Fun and Time2Fun with the model of R\&E [https://aclanthology.org/N18-1044/].  The results are described in the following.

**Experiments to compare R\&E and Word2Fun/Time2Fun, as required by reviewer 6eKb**

The experimental results clearly show that R\&E's model does not perform as well as Word2Fun III. Nevertheless, it could achieve much better results than Time2Fun in time-aware word clustering. Moreover, it achieves comparable effectiveness with Time2Fun in temporal analogy.

In particular, as for time-aware word clustering we found the following results:

|Method |  NMI -10 | $F_\beta$  -10 | NMI -15 | $F_\beta$ -15 | NMI -20 | $F_\beta$ -20 |
|------------------|---------------------------------------|---------|---------| ---- | ---| ---|
| Word2Fun I (Time2Fun) |  0.1703 | 0.1783 | 0.2691 | 0.2680| 0.2842 | 0.2649 |
| Word2Fun III   | **0.7233**|  **0.7080** | **0.7086** | **0.7701** | **0.6980** |  **0.7630**  |
| R \&E      |  0.5726 | 0.5530 | 0.5947| 0.6571 | 0.5877 |0.6883 |



Moreover,  we found the following results for temporal analogy in test$_{1}$:

|Method | MRR | P\@1 | P\@3 | P\@5 | P\@10 |
|--------------|--------------------------|---------|---------| ---- | ---|
| Word2Fun I (Time2Fun) |  0.3735 | 0.2646 | 0.4300 | 0.4955 | 0.5874 |
| Word2Fun III    |  **0.4354** | **0.3076** | **0.5330** | **0.5837** | **0.6647** |
| R \&E      | 0.3357	|0.2638	|0.3493	| 0.4071 | 0.4896 |


Finally, for temporal analogy in test$_{2}$ we found:

|Method | MRR | P\@1 | P\@3 | P\@5 | P\@10 |
|--------------|--------------------------|---------|---------| ---- | ---|
| Word2Fun I (Time2Fun) |  0.0992 | 0.0000 | 0.1315 | 0.1726 | 0.2849 |
| Word2Fun III    | **0.1824** | **0.0795** | **0.1973** | **0.2932** | **0.4164**  |
| R \&E     | 0.0868	|0.0000	| 0.1014	| 0.1425 | 0.2548|

**Why decomposable time-aware word representation (e.g., Time2Fun) performs worse?**

The ineffectiveness of Time2Fun is explained as below. Word/time encoders can be decoupled because of the decomposability property.  Time2Fun can be composed of a word encoder $f$ and a time encoder $g$. In addition, a word $i$ in time $t$ is represented as
$$ U_{i,t} = f_{i} + g_{t} $$
We now considers the two following simple cases:

**Case 1: different words in a specific time/position**

The difference between any two words is always  constant with respect to time:
$$
U_{i,t} - U_{j,t} =   (f_{i} + g_{t}) -  (f_{j} + g_{t}) = f_{i} - f_{j}
$$
Formally, word relationship is independent of time since
$$U_{i,t} - U_{j,t} = \mathcal{E}(i,j) , \quad \forall i$$

This indicates that word relationship does not evolve over time, the latter sounding counter-intuitive; for example, \textit{Trump} and \textit{president} should not be always close in both 2018 and 2021.

**Case 2: the same word in different position/time**

For any word, the difference between its representation in $t_{1}$ and $t_{2}$ is constant.

$$ U_{i,t_{1}} - U_{i,t_{2}} =   (f_{i} + g_{t_{1}}) -  (f_{i} + g_{t_{2}}) = g_{t_{1}} - g_{t_{2}} $$

The result is independent of words.  It follows that all words share the same  evolution  trajectory, namely,
$$
U_{i,t_{1}} - U_{i,t_{2}}  = \mathcal{E} (t_{1},t_{2}) , \quad \forall i
$$

Both cases have contradictions with the reasonable assumptions that (1) individual words may evolve differently, and (2) most words evolve differently in different periods.

**Can  R\&E be decomposed?**

As we explained before,  R\&E without the last output layer can be decomposed as follows:

$$f_{word} (w) =  T\vec{w} + B$$

and

$$ f_{time} (t) = \tanh(M_{2} \tanh(M_{1} t + b_{1})  + b_{2}) $$

A word in time $t$ can be represented as follows:

$$U_{i,t} = f_{word}(w) * f_{time}(t)$$



where $*$ is a matrix-vector product.

The matrix-vector product operation may a better operation than addition in Word2Fun as the interaction module. Moreover, the last output layer of R\&E may help improve the composition of content (i.e., word) and time. This may help account for why R\&E performs much better than Time2Fun with a simple interaction module, i.e., addition.

**A optimization issue of  R\&E, i.e., the gradient explosion**

We also noticed that the training of R\&E is not stable, since it may suffer from gradient explosion, especially when the batch size is small. We also observed that gradient explosion can be reduced by using bigger batch sizes and gradient clip. We suspect that the tensor-vector and matrix-vector product operations may account for gradient explosion, since the results of the operations of R\&E
$$Trans_w = T \vec{w} + B$$
$$h_3 = Trans_w(timevec(t)) $$
$$ use_{W} (w, t) = M_4 h_3 + b_4 $$
are not well-bounded after  tensor-vector and matrix-vector transformation (see the RHS of the above equations). In contrast, sinusoidal functions are bounded in Word2Fun thanks to its periodical properties; dot product in Word2Fun is also bounded thanks to the sigmoid activation.


**Comparison for parameter scale between Word2fun and R\&E**

Let us define $D$ as the dimension of word embedding (i.e, 300) and $d$  as the hidden dimension (i.e, 100) in R\&E.
Therefore, the R\&E model has  $\mathcal{O} (Dd^2 + 2VD + 3d^2 + Dd )$ parameters.
Word2Fun has $\mathcal{O} (kVD)$ parameters. Since $V$ is 20000 or 40000 in the experimental corpora and $k$ is 2-5, we could roughly conclude that the number of parameters of the R\&E model and  that of Word2Fun III are comparable. Time2Fun has fewer parameters.



**The advantages of Word2fun over R\&E**

We would like to highlight that the proposed Word2fun has the following advantages over R\&E.

> - **Theoretical aspect**: In the paper (Sec. 4.1) we discussed the motivations for the use of sinusoidal polynomials and their capability in terms of function approximation because of the Weierstrass Approximation Theorem.

> - **Interpretability**:
The learned functions provide advantages in terms of interpretability over those learned in R\&E. We could use visualization to explicitly show how the learned functions look like.
Note that in Word2Fun each dimension of word embedding is a individual (and also simple) function over time and functions in different dimensions are independent of each other; this simplification makes it possible to visualize learned functions dimension by dimension.
The parameters for the functions like amplitude terms, frequency terms, bias terms, and initial phase terms have concrete physical meanings that could provide us an indication of the speed or of the extent to which the word meaning changes over time. Individual parameters in R\&D are difficult to interpret since many linear/non-linear layers are used and it involves some complicated operations like tensor-vector product (for $Trans_w$ in Eq. 5 of the R\&E paper).

> - **Empirical effectiveness**: Empirical results showed that our model Word2Fun III performs better than R\&E in clustering and semantic analogy.

> - **Better alignment of dynamic word embedding**: The compass that uses a static embedding as target embedding  could help for the alignment. While R\&E should suffer more from the rotation-invariant issues of  word embedding since neither target embedding nor context embedding is  static to time, which might be arbitrarily rotated, as explained in Sec. 2.2 of the paper.




**Code to reproduce the above results**

We will  manage this open-source project, to make the code in the supplementary materials and the R\&E model fully open-sourced, including data, data processing, model design, and evaluation scripts.

The code to reproduce Word2Fun was in the supplementary material. And the R\&E model is reimplemented in https://anonymous.4open.science/r/RosenfeldErkModel-NeurIPS-anoumous-authors/RosenfeldErkModel.py, one can directly plug it in the modeling.py (supplementary material) to reproduce the result of  R\&E in these tables. All models use the same corpora, data processing and optimizer. Thank the timely help from the authors of R\& E, who provide their codes that helps me confirm the correctness of our implementation and redesign the hyperparameters.

---

### Decision · Program_Chairs · 2021-09-27

**Decision:**

Accept (Poster)

**Comment:**

This paper was thoroughly vetted by reviewers who were able to confirm its novelty, limitations (some experimental comparisons would have better contextualised the results), the general importance of the problem (learning diachronic word representations) both in terms of direct applications, and the technical appropriateness of the solution. Like early work championed in ML venues such as LDA that went on to have an important impact in application areas, this is a serious technical contribution that will have an long afterlife in diachronic sociolinguistics.